# Cocaine's cerebrovascular vasoconstriction is associated with astrocytic $Ca^{2+}$ increase in mice

Yanzuo Liu[1], Yueming Hua[1], Kicheon Park [1], Nora D. Volkow [2✉], Yingtian Pan[1] & Congwu Du [1✉]

Human and animal studies have reported widespread reductions in cerebral blood flow associated with chronic cocaine exposures. However, the molecular and cellular mechanisms underlying cerebral blood flow reductions are not well understood. Here, by combining a multimodal imaging platform with a genetically encoded calcium indicator, we simultaneously measured the effects of acute cocaine on neuronal and astrocytic activity, tissue oxygenation, hemodynamics and vascular diameter changes in the mouse cerebral cortex. Our results showed that cocaine constricted blood vessels (measured by vessel diameter $\Phi$ changes), decreasing cerebral total blood volume (HbT) and temporally reducing tissue oxygenation. Cellular imaging showed that the mean astrocytic $Ca^{2+}$ dependent fluorescence ($\Delta F/F_{Ca^{2+}-G(A)}$) increase in response to cocaine was weaker but longer lasting than the mean neuronal $Ca^{2+}$ dependent fluorescence ($\Delta F/F_{Ca^{2+}-G(N)}$) changes. Interestingly, while cocaine-induced $\Delta F/F_{Ca^{2+}-G(N)}$ increase was temporally correlated with tissue oxygenation change, the $\Delta F/F_{Ca^{2+}-G(A)}$ elevation after cocaine was in temporal correspondence with the long-lasting decrease in arterial blood volumes. To determine whether the temporal association between astrocytic activation and cocaine induced vasoconstriction reflected a causal association we inhibited astrocytic $Ca^{2+}$ using GFAP-DREADD(Gi). Inhibition of astrocytes attenuated the vasoconstriction resulting from cocaine, providing evidence that astrocytes play a critical role in cocaine's vasoconstrictive effects in the brain. These results indicate that neurons and astrocytes play different roles in mediating neurovascular coupling in response to cocaine. Our findings implicate neuronal activation as the main driver of the short-lasting reduction in tissue oxygenation and astrocyte long-lasting activation as the driver of the persistent vasoconstriction with cocaine. Understanding the cellular and vascular interaction induced by cocaine will be helpful for future putative treatments to reduce cerebrovascular pathology from cocaine use.

[1] Department of Biomedical Engineering, Stony Brook University, Stony Brook, NY 11794, USA. [2] National Institute on Drug Abuse, Bethesda, MD 20852, USA.
✉email: nvolkow@nida.nih.gov; Congwu.Du@stonybrook.edu

Cocaine is a highly rewarding and addictive drug[1], whose misuse is associated with significant morbidity and mortality. Among the most serious adverse effects from cocaine are cerebrovascular complications that result in transient ischemia and strokes[2–4]. Indeed, numerous clinical brain imaging studies have reported widespread reductions in cerebral blood flow and cerebral blood volume in individuals with cocaine use disorder[5,6], and preclinical optical imaging studies have reported similar findings in rodent models[7–9]. However, the molecular and cellular mechanisms underlying cerebral blood flow reductions are not well understood and could reflect (1) direct vasoconstrictive properties of cocaine in blood vessels and/or indirect vasoconstriction secondary to release of sympathomimetic amines[10,11], or (2) neuronal deficits that decrease flow secondary to reduced neuronal activity[12] and metabolic demand[13,14], or (3) astrocyte deficits that impair cerebral blood flow homeostasis[15–17].

Neurovascular coupling is involved in modulation of brain function[18]. Neuronal-vascular interactions are necessary to maintain an adequate supply of oxygen and glucose for proper neuronal function[19,20] and until recently most studies on neurovascular coupling focused on neurons. However, there is now increasing interest to study the interactions between neurons and glial cells and their role in neurovascular coupling. Astrocytes provide a cellular link between neuronal circuitry and blood vessels[19–21], astrocytic processes wrap around neuronal synapses while astrocytic feet wrap around blood vessels[22]. This linkage suggests that astrocytes are not only essential for the supply of energy to neurons but can also transfer signals that regulate blood flow in response to neuronal activity[23,24].

Astrocytes provide structural and nutritional support for neurons and participate in the maintenance of the blood brain barrier[25,26]. They also support brain function through their regulation of cerebral blood flow, neuromodulation, and balancing synaptic transmission[27,28]. How astrocytes respond to cocaine, which profoundly affects neurons[29,30] and cerebral blood vessels[17,31] is still not clear. It is likely that cocaine's effects on brain function reflect the interaction between vascular systems, neurons and astrocytes, which may change as a function of time from cocaine exposure and chronicity. Distinguishing these effects would ideally require simultaneous multi parameter longitudinal measurements in vivo. In addition, studying the roles of astrocytes in brain function is difficult because they are essential for neuronal survival and their removal causes neuronal death. Thus, much of what we know about astrocyte function is derived from studies of isolated mammalian astrocytes in vitro[27,32], which cannot inform us on how astrocytes interact with neurons and the surrounding vessels.

To address these challenges, in this study we simultaneously measured the effects of cocaine on neuronal or astrocytic activity, tissue oxygenation, vascular hemodynamics, and vascular diameter changes in the mouse cerebral cortex. To do so, we used a viral vector to express a genetically encoded $Ca^{2+}$ indicator, GCaMP6f, into neurons[33] or astrocytes[34] within the somatosensory cortex of mice[35]. Our multimodality imaging platform through a cranial window[36] enabled us to image cell-specific $Ca^{2+}$ dependent fluorescence changes from astrocytes or neurons while concurrently measuring dynamic changes in cerebral blood volume (i.e., total hemoglobin concentration [HbT]) and tissue oxygenation with vascular changes separately for veins and arteries at high spatiotemporal resolution over a relatively large field of view (FOV). Meanwhile, we used a custom Matlab program to quantify changes in vessel diameter sizes after cocaine to provide a direct measurement of vascular responses. We show that the mean astrocytic Ca dependent fluorescence ($\Delta F/F_{Ca^{2+}\text{-}G(A)}$) response to cocaine was weaker, and longer

lasting than the mean neuronal Ca dependent fluorescence ($\Delta F/F_{Ca^{2+}\text{-}G(N)}$) responses. Interestingly, while cocaine-induced $\Delta F/F_{Ca^{2+}\text{-}G(N)}$ increases were temporally correlated with tissue oxygenation changes, the $\Delta F/F_{Ca^{2+}\text{-}G(A)}$ elevation after cocaine was in temporal correspondence with the long-lasting decrease in HbT in arteries. To evaluate whether the temporal association between astrocytic activation and cocaine induced vasoconstriction reflects a causal association, we used GFAP-DREADD(Gi) to inhibit astrocytic $Ca^{2+}$-G(A). Inhibition of astrocytes attenuated the vasoconstriction from cocaine, thus providing evidence that astrocytes play a critical role underlying cocaine's vasoconstrictive effects in cerebral blood vessels.

## Results

### GCaMP6f-expressed $Ca^{2+}$ Fluorescence in Astrocytes/Neurons.
To track neuronal or astrocytic activation in response to cocaine, we delivered GCaMP6f, a genetically encoded $Ca^{2+}$ indicator, into the cortex of C57BL/6 wild type and GFAP-Cre mice (Fig. 1c), respectively. For example, Fig. 1b illustrates the astrocytic GCaMP6f $Ca^{2+}$ fluorescence image (i) along with the hemodynamic images (ii, iii) in the mouse cortex acquired with the multimodality imaging platform in vivo. To corroborate that GCaMP6f was specifically expressed in astrocytes or neurons, we used immunohistochemistry to label either neurons with NeuN or astrocytes with GFAP antibodies for brain sections from wild type and GFAP-Cre animals, respectively. Figure 1d shows representative ex vivo coronal sections of cortex with GCaMP6f-expressing neurons (i) and astrocytes (ii) visualized by anti-GFP staining in wild type and GFAP-Cre mice, respectively. Their 'zoom-in' images within the white boxes in Fig. 1d show that GFP + cells were all neurons in wild-type mice (iii) and astrocytes in GFAP-Cre mice (iv). These ex vivo images confirmed that GCaMP6f fluorescence in wild type mice was from neuronal $Ca^{2+}$ and that in GFAP-Cre mice was from astrocytic $Ca^{2+}$. Quantification in Supplementary Fig. 1j shows that levels of GCaMP6f expression in neurons (55.6% ± 8.9%, $n = 5$, wild type mice) and in astrocytes (56.4% ± 4.5%, $n = 5$, GFAP-Cre mice) did not differ ($p = 0.94$), indicating similar cortical $Ca^{2+}$ dependent fluorescence signaling from neurons in wild type mice and from astrocytes in GFAP-Cre mice.

### Vascular [HbT] and [ΔΦ] decrease due to cocaine-induced vasoconstriction.
Figure 2 summarizes cocaine's effects on the cortical vasculature including changes in total hemoglobin ($\Delta$[HbT]) and in vascular diameter ($\Delta\Phi$) in veins and arteries. $\Delta$[HbT] and $\Delta\Phi$ represent the changes in total blood volume and vessel morphology, respectively, due to vasoconstriction from cocaine. Both wild type and GFAP-Cre mice were included, and their vascular changes indicated as $\Delta$[HbT-G(N)], $\Delta$[Φ-G(N)] and $\Delta$[HbT-G(A)], $\Delta$[Φ-G(A)], respectively. Figure 2a shows the absorption spectra of oxygenated- (HbO$_2$, red curve) and deoxygenated- (HbR, blue curve) hemoglobin at different wavelengths, which can be used to calculate HbO$_2$ and HbR (Eq.1), and to estimate HbT within vessels and tissue (Eq. 2). This is because $\lambda_1 = 568$ nm is an isosbestic point of HbO$_2$ and HbR spectra with high absorbance, enabling imaging blood absorption in both arteries (HbO$_2$ dominant) and veins (HbR dominant). The specific wavelengths of $\lambda_1 = 568$ nm and $\lambda_2 = 630$ nm were used for imaging since at $\lambda_2 = 630$ nm HbR absorption is higher than HbO$_2$, which allows to distinguish veins from arteries in the cortex of mice from either HbT-G(N) (i neuronal) or HbT-G(A) (ii astrocytic) groups (Fig. 2b). Regions of interest (ROIs) were selected from arteries (e.g., red traces) and veins (e.g., blue traces) for each animal and their temporal changes with cocaine

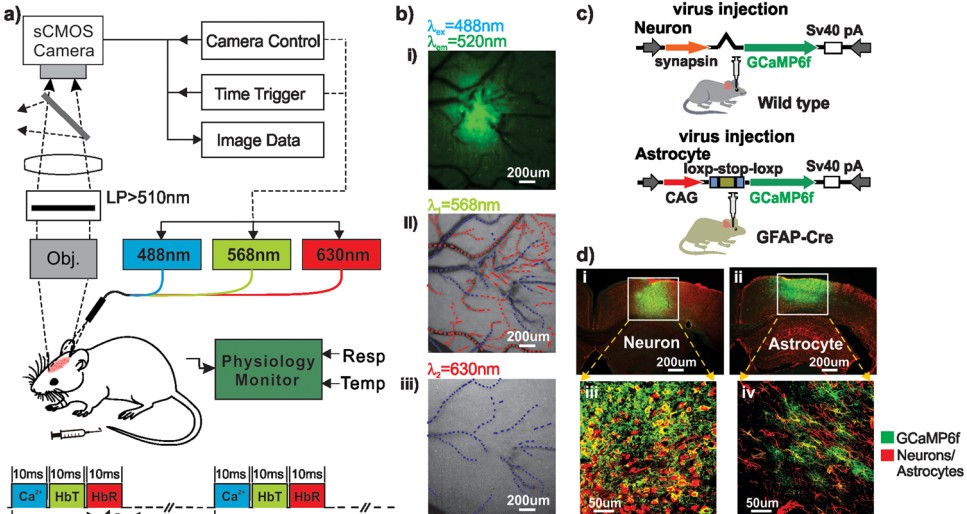

**Fig. 1 A schematic illustration of the experimental design and animal models. a** Multimodality imaging platform allows simultaneous imaging of cellular $Ca^{2+}$ dependent fluorescence and hemodynamic changes in response to cocaine. LP: Long-pass filter >510 nm. **b** Cortical images acquired with the different channels including fluorescence image (i: $\lambda_{ex} = 488$ nm, $\lambda_{em} = 520$ nm), reflected images at $\lambda_1 = 568$ nm (ii) and $\lambda_2 = 630$ nm (iii) to separate arteries (red tracks) from veins (blue tracks) and to retrieve hemodynamics ($HbO_2$ and $HbR$) in brain tissue. **c** Animal models for viral delivery of GCaMP6f to neurons and astrocytes to mouse somatosensory cortex in vivo. **d** Ex-vivo confocal fluorescence images to show GCaMP6f distribution in cortical neurons (i) and astrocytes (ii), respectively. Zoomed-in views to confirm GCaMP6f-expressions in neurons (iii) and astrocytes (iv) within the white boxes. Green: GCaMP6f-expressing neurons and Red: NeuN (iii), Green: GCaMP6f-expressing astrocytes and Red: GFAP stains (iv) to represent neurons and astrocytes, respectively.

extracted. Figure 2c, d show $\Delta[HbT\text{-}G(N)]$ and $\Delta[HbT\text{-}G(A)]$ changes with time after cocaine (1 mg/kg, i.v.), respectively, which reveals that total blood volumes in arteries and veins decreased immediately ($\Delta t = 2$ min) after cocaine to $\sim -20\%$ ($t = 5$–10 min) below their baseline followed by gradual recovery. Meanwhile, Fig. 2e, f show $\Delta[\Phi\text{-}G(N)]$ and $\Delta[\Phi\text{-}G(A)]$ changes as a function of time from cocaine injection (1 mg/kg, i.v.), respectively, which indicate that vessel sizes of arteries and veins decreased to $\sim -10\%$ ($t = 0$–20 min) and slowly returned to baseline within $t = 60$ mins. The mean $\Delta[HbT]$ decay rates ($\Delta[HbT]\%/min$) are summarized in Fig. 2g for neuronal- (Group N) and astrocytic- GCaMP6f expressing (Group A) mice. For Group N, $\Delta[HbT\text{-}G(N)]\%/min$ in arteries and veins were $-15.66 \pm 2.464\%/min$ and $-12.26 \pm 2.491\%/min$, respectively; for Group A, $\Delta[HbT\text{-}G(A)]\%/min$ in arteries and veins were $-17.15 \pm 5.041\%/min$ and $-16.68 \pm 5.351\%/min$, respectively. The arteries' changes from cocaine did not differ between Group N and Group A ($p = 0.992$), which indicates that viral GCaMP6f delivery did not affect vascular reactivity to cocaine. Comparison of cocaine-induced integrative $\Delta[\Phi]$ changes in Group A and Group N is summarized in Fig. 2h. For Group N, $\Delta[\Phi\text{-}G(N)]\%/min$ in arteries and veins were $-5.544 \pm 1.210\%min$ and $-6.818 \pm 2.110\%min$; for Group A, $\Delta[\Phi\text{-}G(A)]\%min$ in arteries and veins were $-6.437 \pm 1.702\%/min$ and $-5.402 \pm 2.520\%min$. Cocaine induced changes in integrative $\Delta[\Phi]$ did not differ between Group A and Group N nor between arteries and veins.

**Cocaine induces a transient decrease in tissue oxygenation.** Figure 3 shows cocaine-induced changes in tissue oxygen hemoglobin ($HbO_2$), i.e., $[HbO_2]\text{-}G(N)$ and $[HbO_2]\text{-}G(A)$ in the cortices of Group N (neuronal) and Group A (astrocytic) mice. Figure 3a, b show representative baseline raw image at $\lambda_1 = 568$ nm and time-lapse $\Delta[HbO_2]$ ratio images after cocaine (1 mg/kg, i.v., $t = 0$ min). Figure 3c show the time courses of mean $\Delta[HbO_2]$ changes after cocaine in Group N (solid line, ROIs=3/animal, $n = 5$) and Group A (dashed line, ROIs=3/animal, $n = 5$) mice. Cocaine immediately reduced $\Delta[HbO_2]$ to $-12.09 \pm 3.777\%$

(Group N) and $-9.467 \pm 2.612\%$ (Group A) at $\sim 10$ min after cocaine that gradually recovered at $24.2 \pm 3.625$ min in Group N and at $25 \pm 5.692$ min in Group A, followed by $\sim 10$–15% overshoot over baseline at $t = 60$ min. Comparisons of peak decrease and reduction duration between Group N and Group A showed no differences (Fig. 3d, e), indicating that viral GCaMP delivery did not affect cortical tissue function in response to cocaine.

**Cocaine induces a robust transient increase in neuronal $Ca^{2+}$ but a weak and persistent increase in astrocytic $Ca^{2+}$.** Figure 4 shows $Ca^{2+}$ dependent fluorescence changes in neurons ($\Delta F/F_{Ca^{2+}\text{-}G(N)}$) and astrocytes ($\Delta F/F_{Ca^{2+}\text{-}G(A)}$) in response to cocaine (1 mg/kg, i.v). The spatiotemporal evolutions of cocaine-induced $\Delta F/F_{Ca^{2+}\text{-}G(N)}$ and $\Delta F/F_{Ca^{2+}\text{-}G(A)}$ at $t = 0, 5, 25$, and 40 min are illustrated in Fig. 4a, b, respectively; their mean increases with time for Group N ($n = 5$) and Group A ($n = 5$) mice are plotted in Fig. 4c. Both $\Delta F/F_{Ca^{2+}\text{-}G(N)}$ and $\Delta F/F_{Ca^{2+}\text{-}G(A)}$ fluorescence responded promptly to a cocaine injection. However, the neuronal reaction $\Delta F/F_{Ca^{2+}\text{-}G(N)}$ was larger (>6% increase over baseline) and recovered within 30–40 min with a downshoot afterwards, whereas the astrocytic reaction in $\Delta F/F_{Ca^{2+}\text{-}G(A)}$ was weaker, only 2%, but was longer lasting (>50 min) after cocaine. The peak increases following cocaine injection of $6.065 \pm 1.463\%$ for $\Delta F/F_{Ca^{2+}\text{-}G(N)}$ ($n = 5$, ROIs = 5/animal) were significantly higher ($p^* = 0.032$) than $2.106 \pm 0.4333\%$ for $\Delta F/F_{Ca^{2+}\text{-}G(A)}$ ($n = 5$, ROIs = 5/animal) (Fig. 4d). Their recovery time of $32.2 \pm 1.933$ min for $\Delta F/F_{Ca^{2+}\text{-}G(N)}$ was significantly faster than $54.6 \pm 4.167$ min for $\Delta F/F_{Ca^{2+}\text{-}G(A)}$ (Fig. 4e), indicative of longer-lasting cocaine effects on astrocytic than neuronal $Ca^{2+}$ signaling ($p^* < 0.001$).

**Cocaine-induced Neuronal $\Delta F/F_{Ca^{2+}\text{-}G(N)}$ fluorescent change correlates with tissue $\Delta HbO_2$.** Figure 5a shows simultaneous changes in neuronal $Ca^{2+}$ dependent fluorescence ($\Delta F/F_{Ca^{2+}\text{-}G(N)}$) and oxygenated-hemoglobin ($\Delta HbO_2\text{-}G(N)$)

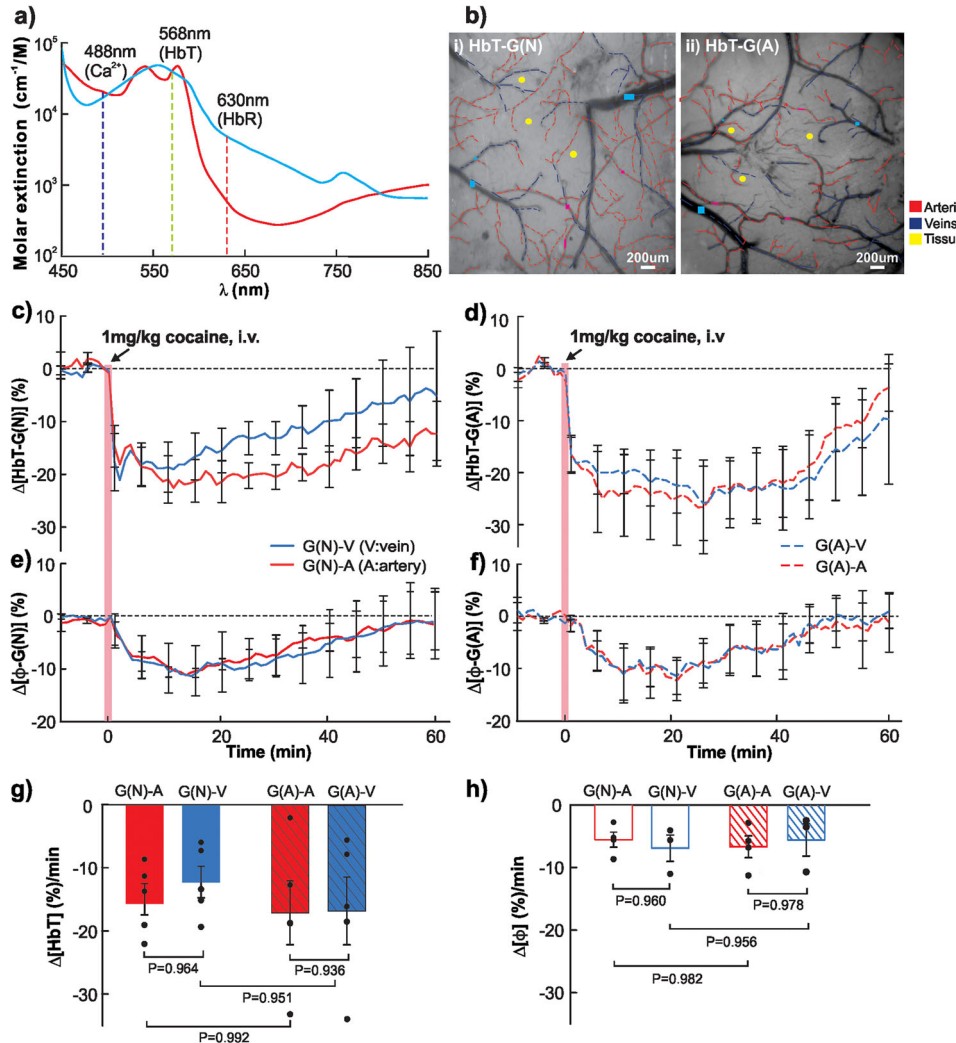

**Fig. 2 Cocaine-induced total hemoglobin (i.e., blood volume) and vessel diameter changes in veins and arteries of mouse brains in vivo. a** Absorption spectra of oxygenated-hemoglobin (HbO₂, red curve) and deoxygenated-hemoglobin (HbR, blue curve). Besides $\lambda_{ex} = 488$ nm for excitation of cellular GCaMP6f Ca²⁺ fluorescence, $\lambda_1 = 568$ nm and $\lambda_2 = 630$ nm were selected in multimodality imaging platform to retrieve changes in HbO₂ and HbR, thus HbT in cortical tissue as well as within vessels. **b** Spectral images of the cortex at $\lambda_1 = 568$ nm in WT mouse (i) and GFAP-Cre mouse (ii) with separation of arteries (red traces) and veins (blue traces). Regions of interest (ROIs) were selected from arteries (e.g., pink dots), veins (e.g., blue dots) and tissue (e.g., yellow dots) from each animal. **c, d** Time courses of Δ[HbT] in arteries (red) and veins (blue) in response to cocaine (1 mg/kg, i.v.) in neuronal (**c**) and astrocytic (**d**) GCaMP6f-expressing animals (*n* = 5/group), both showing long-persistent decreases in arteries and veins by cocaine. **e, f** Vascular diameter (ΔΦ) changes in arteries (red) and veins (blue) as a function of time after cocaine injection (1 mg/kg, i.v.) in neuronal and astrocytic groups, respectively. **g** Comparisons of Δ[HbT]% decrease per minute between arteries and veins in the neuronal and astrocytic GCaMP6f-expressing animals (*n* = 5/group), showing no significant difference between arteries and veins in either group. **h** Comparisons of ΔΦ% decrease per minute between arteries and veins in neuronal and astrocytic GCaMP-expressing groups, showing no significant difference of ΔΦ% in response to cocaine. All error bars are presented as means ± SEM.

within cortical tissue and blood volume (ΔHbT-G(N)) in arteries before and after cocaine (Group N, *n* = 5). $\Delta F/F_{\text{Ca}^{2+}\text{-G(N)}}$ increased 6.065 ± 1.463% at 8–10 min after cocaine recovering to its baseline value at 32.2 ± 1.933 min followed by a downshoot till the end of recording (*t* = 60 min, green curve). Meanwhile, ΔHbO₂ decreased −12.09 ± 3.777% at 8–10 min after cocaine and recovered at 24.2 ± 3.625 min followed by an overshoot (orange curve). Comparison of recovery time to the baseline between ΔCa²⁺-G(N) of 32.2 ± 1.93 min and Δ[HbO₂]-G(N) of 24.2 ± 3.625 min revealed no difference (*p* = 0.087). In contrast, Δ[HbT]-G(N)) in arteries decreased −22.56 ± 4.496% at ~10 min after cocaine, remained low and then slightly recovered to −12.31 ± 6.122% at *t* = 60 min after cocaine (red curve).

The correlation analysis between cocaine-induced $\Delta F/F_{\text{Ca}^{2+}\text{-G(N)}}$ and Δ[HbO₂]-G(N) for the time periods of peak response (*t* = 0–30 min, 'cocaine quadrant region') and overshoot (*t* = 30–60 min, 'overshoot quadrant region') is shown in Fig. 5b. The linear regression plot revealed a strong inverse correlation between neuronal Ca²⁺ and tissue ΔHbO₂ (*r* = −0.984, *p* < 0.01), where colored dots represent different time periods from *t* = −10 min baseline to *t* = 60 min after cocaine. Though our findings cannot establish causality they suggest that cocaine-induced reduction in tissue oxygen content in the peak response period might reflect increased neuronal activation, whereas the increase in tissue oxygen content that followed might reflect decreased neuronal activity during the overshoot. In contrast, cocaine induced neuronal Ca²⁺

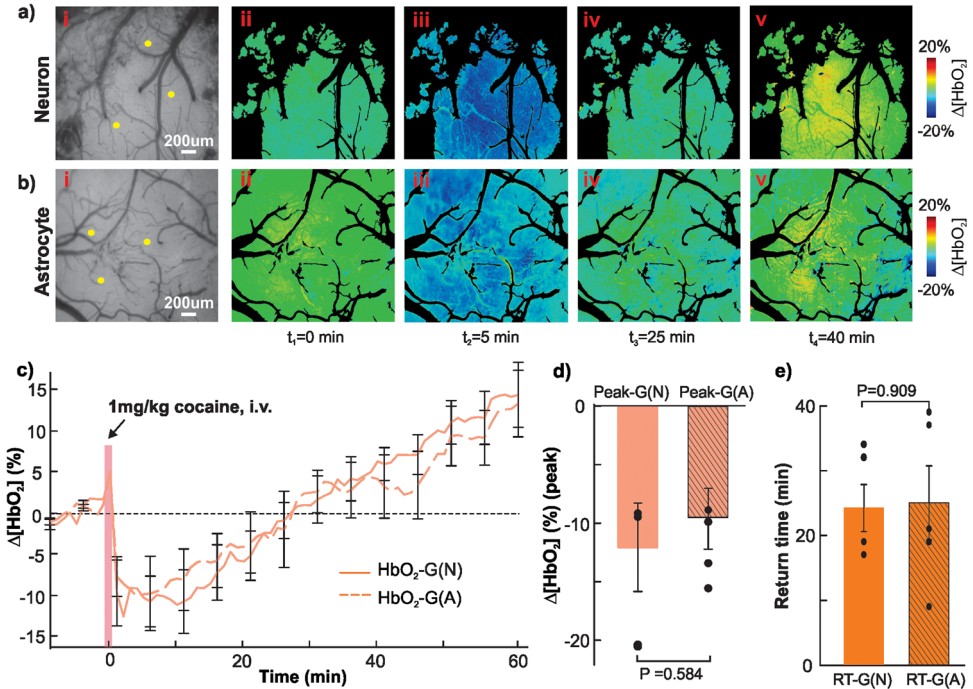

**Fig. 3 Comparison of cocaine-induced tissue oxygenated-hemoglobin (HbO₂) changes between Group N (neuronal) and Group A (astrocytic) mice.** **a**, **b** Time lapse Δ[HbO₂] images in response to cocaine (1 mg/kg, i.v) in a Group N animal (ii–v) and a Group A animal (ii–v) at $t = 0, 5, 25, 40$ min, respectively. Three ROIs were selected in tissue (e.g., yellow dots) from each animal. **c** Mean Δ[HbO₂] changes with time in response to cocaine (1 mg/kg, i.v) in Group N (solid line, $n = 5$) and Group A (dashed line, $n = 5$), showing cocaine-induced Δ[HbO₂] decrease in cortical tissue over 20–25 min followed by an overshoot over baseline about 10–15% at $t = 60$ min after cocaine. **d** Comparison of peak Δ[HbO₂] decreases between G(N) and G(A), showing no difference ($n = 5$, $p = 0.584$). **e** Comparison of Δ[HbO₂] return time to baseline between G(N) and G(A), showing no difference ($n = 5$, $p = 0.909$). All error bars are presented as means ± SEM.

did not correlate with arterial ΔHbT (Supplementary Fig. 2), indicating that arterial volume changes with cocaine were not driven by changes in neuronal activity.

**Cocaine-induced astrocytic $\Delta F/F_{Ca^{2+}\text{-}G(A)}$ fluorescent change correlates with HbT in arteries.** Cocaine-induced simultaneous changes in astrocytic Ca²⁺ ($\Delta F/F_{Ca^{2+}\text{-}G(A)}$) dependent fluorescence and oxygenated-hemoglobin (Δ[HbO₂]-G(A)) within cortical tissue and Δ[HbT]-G(A) in arteries ($n = 5$) showed $\Delta F/F_{Ca^{2+}\text{-}G(A)}$ increased $2.106 \pm 0.4333\%$ at $t \approx 3$ min after cocaine and plateaued till $t \approx 18$ min followed by a gradual slow recovery till $t \approx 60$ min (green curve in Fig. 6a). Changes in Δ[HbT]-G(A) showed a similar temporal pattern to that of $\Delta F/F_{Ca^{2+}\text{-}G(A)}$, which decreased $-24.96 \pm 8.415\%$ at $t = 2$ min, remained largely unchanged till $t = 35$ min followed by a slow recovery to $-3.634 \pm 4.544\%$ at $t = 60$ min after cocaine (red curve). In contrast, Δ[HbO₂]-G(A) decreased $-9.467 \pm 2.612\%$ after cocaine and recovered at $t = 25 \pm 5.692$ min with an overshoot (orange curve).

Correlation analysis between the temporal changes in astrocytic $\Delta F/F_{Ca^{2+}\text{-}G(A)}$ and Δ[HbT]-G(A) in arteries from baseline ($t = -10–0$ min) to 60 min after cocaine showed a strong inverse correlation ($R = -0.786$, $p < 0.01$) as shown in Fig. 6b, where colored dots represent different time periods of recording. This finding suggests that astrocytic activation by cocaine underlie cocaine induced vasoconstriction. In contrast, there was no association between $\Delta F/F_{Ca^{2+}\text{-}G(A)}$ and tissue Δ[HbO₂]-G(A) (Supplementary Fig. 3).

**Inhibition of astrocytic fluorescence $\Delta F/F_{Ca^{2+}\text{-}G(D)}$ activity with GFAP-DREADD (Gi) reduces cocaine's vasoconstriction.** Figure 7 shows the comparison of astrocytic $\Delta F/F_{Ca^{2+}\text{-}G(D)}$, vessels'

diameter and hemodynamic responses to cocaine in cortex of mice ($n = 5$) before and after DREADD(Gi) activation by clozapine. For ΔHbT, before DREADD (Gi) activation, cocaine immediately reduced ΔHbT to $-18.84 \pm 1.115\%$ which did not return to baseline over 60 mins after cocaine. However, after 30 mins of clozapine's activation of DREADD, cocaine-induced decrease in ΔHbT were much smaller ($p < 0.05$, maximum of $-5.514 \pm 0.5085\%$ at $t = 2–8$ min) and rapidly returned to baseline after 8 min ($p > 0.05$). Vessel diameter size (ΔΦ) in response to cocaine, before clozapine decreased to $-7.557 \pm 2.463\%$ gradually recovering to baseline at $t = 60$ min, whereas after clozapine there was no significant changes in Δφ compared to baseline throughout the whole experiment time period (Fig. 7f). Persistent Δ [HbT] (Fig. 7c) and Δφ (Fig. 7e) decreases reflected cocaine's vasoconstriction effects along with the corresponding [Ca²⁺]-G(D) fluorescent increase ($\Delta F/F_{Ca^{2+}\text{-}G(D)}$), Fig. 7a. After astrocytic $\Delta F/F_{Ca^{2+}\text{-}G(D)}$ inhibition, $\Delta F/F_{Ca^{2+}\text{-}G(D)}$ was slightly increased (<1%) after cocaine for a short time period (2–5 min; Fig. 7b). Tissue ΔHbO₂ after cocaine before clozapine, was reduced to $-11.81 \pm 1.047\%$ and gradually recovered to baseline at $30.4 \pm 8.853$ min followed by an overshoot, consisted with observation shown in Fig. 3 above. After clozapine, tissue ΔHbO₂ following cocaine injection was mildly reduced $-5.118 \pm 1.609\%$ within $t = 2–7$ mins, with a fast recovery after 10 min.

Comparisons of cocaine-induced mean fluorescent changes in $\Delta F/F_{Ca^{2+}\text{-}G(D)}$, ΔHbT, ΔHbO₂, and Δ Φ before and after DREADD(Gi) activation by clozapine are summarized in Fig. 7i–l, showing that the cocaine-induced changes in astrocytic [Ca²⁺]-G(D) fluorescence, ΔHbT, Δφ, and ΔHbO₂, are significantly reduced after astrocytic ΔCa²⁺ inhibition. Specifically, without or with astrocytic ΔCa²⁺ inhibition, the mean $\Delta F/F_{Ca^{2+}\text{-}G(D)}$ changed from $0.5046 \pm 0.1169\%$/min to $-0.1672 \pm 0.1180\%$/min ($p^* = 0.004$);

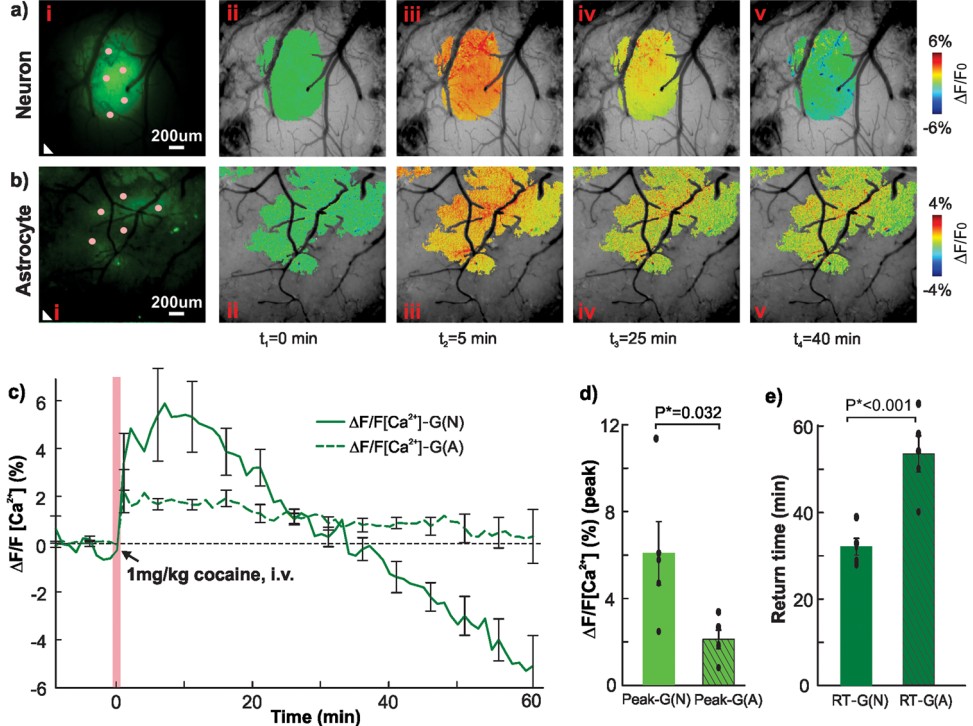

**Fig. 4 Cocaine-induced cellular Ca$^{2+}$ dependent fluorescence changes in neuronal ($\Delta F/F_{Ca^{2+}-G(N)}$) and astrocytic ($\Delta F/F_{Ca^{2+}-G(A)}$) mice. a, b** Time lapse images of Ca$^{2+}$ fluorescence ratio changes $\Delta F/F_{Ca^{2+}-G(N)}$ (ii-v) and $\Delta F/F_{Ca^{2+}-G(A)}$ (ii-v) in response to cocaine (1 mg/kg, i.v) superposed on their baseline spectral images (i) obtained at $\lambda_1 = 568$ nm. Five Regions of interest (ROIs) were selected from fluorescence expressing regions (e.g., pink dots) and one ROI from outside expressing area (e.g., white triangle), that was used to correct for confounding artifacts. **c** Cocaine-induced mean Ca$^{2+}$ fluorescence changes in neurons ($\Delta F/F_{Ca^{2+}-G(N)}$, solid line, $n = 5$), and astrocytes ($\Delta F/F_{Ca^{2+}-G(A)}$ dash line, $n = 5$), both showing immediate $\Delta F/FCa^{2+}$ increases after cocaine (1 mg/kg, i.v). The $\Delta F/F_{Ca^{2+}-G(N)}$ increase was robust and returned to baseline within ~30 min, whereas the $\Delta F/F_{Ca^{2+}-G(A)}$ increase was smaller (~2%) but longer lasting. **d** Comparison of peak increases in $\Delta F/F_{Ca^{2+}-G(N)}$ vs $\Delta F/F_{Ca^{2+}-G(A)}$, show that the astrocytic response to cocaine was lower than the neuronal response ($p^*=0.032$). **e** Comparison of recovery time between $\Delta F/F_{Ca^{2+}-G(N)}$ and $\Delta F/F_{Ca^{2+}-G(A)}$ indicated that the astrocytic response was longer lasting than the neuronal response ($p^*<0.001$). All error bars are presented as means ± SEM.

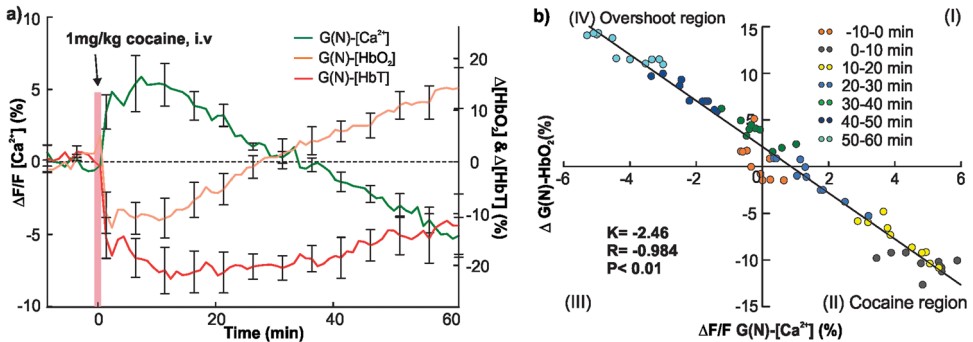

**Fig. 5 Comparison of cocaine's effects on neuronal fluorescence $\Delta F/F_{Ca^{2+}-G(N)}$, tissue $\Delta[HbO_2]$, and $\Delta[HbT]$ in arteries. a** Mean $\Delta F/F_{Ca^{2+}-G(N)}$ (green), $\Delta[HbO_2]$-G(N) (orange), and $\Delta[HbT]$-G(N) (red) changes with time in cortex in response to cocaine (1 mg/kg, i.v.) ($n = 5$). **b** Correlation between cocaine-induced response in $\Delta F/F_{Ca^{2+}-G(N)}$ and $\Delta[HbO_2]$-G(N) showed a strong inverse linear association ($R = -0.984$, $p < 0.01$). All error bars are presented as means ± SEM.

$\Delta$HbT changed from $-9.668 \pm 0.4799\%$/min to $-0.6927 \pm 0.0933\%$/min ($P^* < 0.001$); $\Delta\phi$ decreased from $-3.413 \pm 0.5230\%$/min to $-0.4476 \pm 0.3379\%$/min ($P^* = 0.009$); and $\Delta$HbO$_2$ from $-10.482 \pm 3.8785\%$/min to $-2.179 \pm 0.3419$ %/min, ($P^* = 0.048$), respectively.

To determine whether clozapine would affect fluorescent neuronal Ca$^{2+}$-G(C), and vessel diameters in the cortex, we administered clozapine (0.1 mg/kg, i.p.) to a group of animals ($n = 3$, shown in Table 1) during experiments and recorded changes in neuronal Ca$^{2+}$ fluorescence ($\Delta F/F_{Ca^{2+}-G(C)}$) and vessel diameter size in at baseline (10 mins) and 30 mins after clozapine injection. Results are summarized in Supplementary Fig. 4. Supplementary Fig. 4a shows neuronal $\Delta F/F_{Ca^{2+}-G(C)}$ changes as a function of time in response to clozapine. One-way repeated ANOVA showed a no significant time effect on neuronal $\Delta F/F_{Ca^{2+}-G(C)}$ after clozapine injection. Quantification analysis of average efficiency in Supplementary Fig. 4b shows no significant differences before ($0.2145 \pm 0.1361\%$) and after ($0.3940 \pm 0.0940\%$) clozapine injection ($p = 0.339$). Supplementary Figure 4c shows the time traces of the vessel diameter change

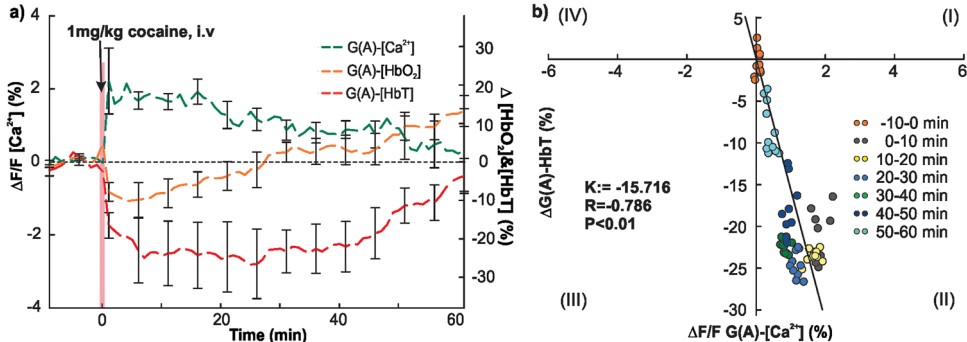

**Fig. 6 Comparison of cocaine's effects on astrocytic fluorescence $\Delta F/F_{Ca^{2+}\text{-G(A)}}$, tissue $\Delta[HbO_2]$, and $\Delta[HbT]$ in arteries. a** Mean $\Delta F/F_{Ca^{2+}\text{-G(A)}}$ (green), $\Delta[HbO_2]$-G(A) (orange), and $\Delta[HbT]$-G(A) (red) changes with time in cortex in response to cocaine (1 mg/kg, i.v.) ($n = 5$). **b** Correlation analysis between cocaine-induced response in astrocytic $\Delta F/F_{Ca^{2+}\text{-G(A)}}$ and $\Delta[HbT]$ in arteries, showing a strong inverse correlation ($R = -0.786$, $p < 0.01$). All error bars are presented as means ± SEM.

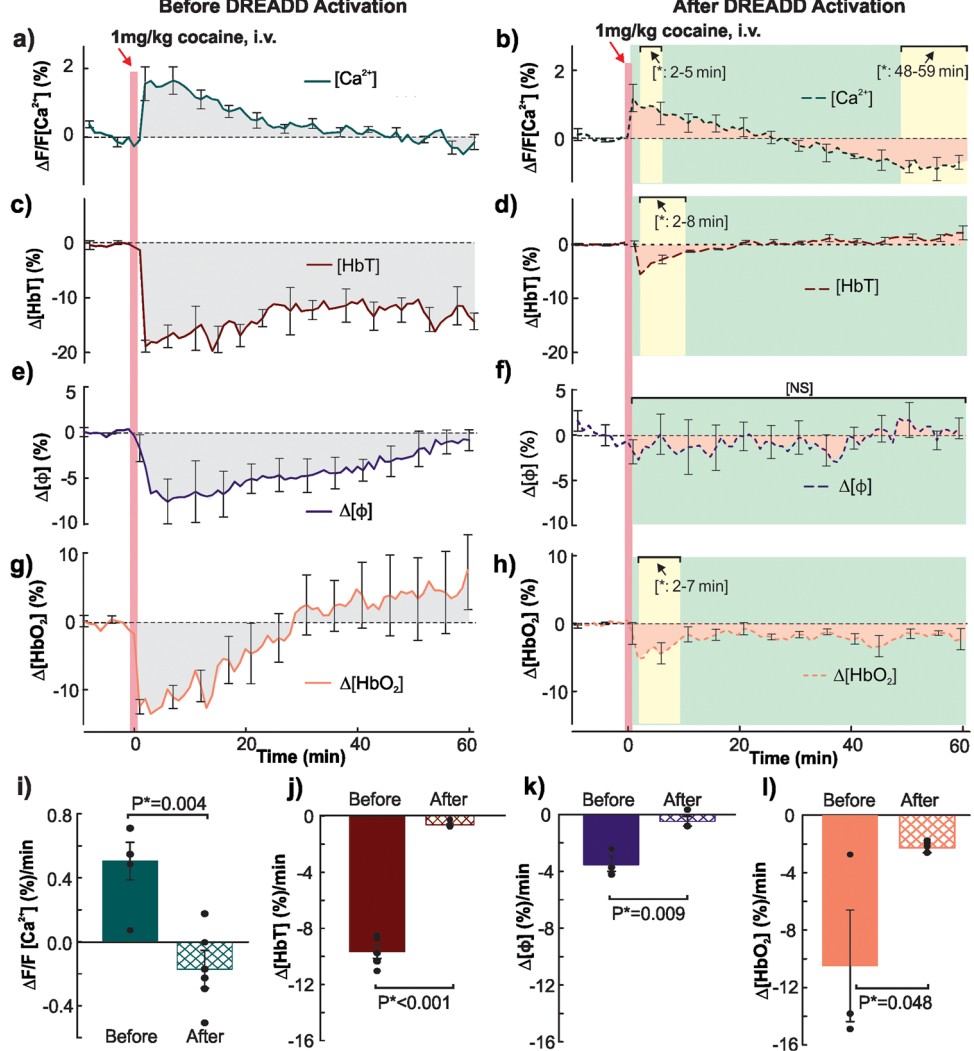

**Fig. 7 Inhibiting astrocytic activity with clozapine in GFAP-DREADD (Gi) expressing mice blocked cocaine-induced vasoconstriction.** Cocaine induced $\Delta F/F_{Ca^{2+}\text{-G(D)}}$ (**a**, **b**), HbT (**c**, **d**), vessel diameter size (**e**, **f**), and $HbO_2$ (**g**, **h**) changes as function of time before and after astrocytic $\Delta Ca^{2+}$ inhibition by clozapine. **i–l** Quantification analysis of average efficiency in $\Delta F/F_{Ca^{2+}\text{-G(D)}}$, vessels size, hemodynamic response respectively before DREADD(Gi) activation (Gray shadow in **a**, **c**, **e**, **g**) and after DREADD(Gi) activation by clozapine (Pink shadow in **b**, **d**, **f**, **h**). Meanwhile, yellow shadow indicates time periods with significant differences compared to baseline ($p < 0.05$), and green shadow indicate no-significant differences ($p > 0.05$). All error bars are presented as means ± SEM.

**Table 1 Animal groups and experimental design.**

| Experiment | Pre-virus injection | Drug challenge | Detection/ imaging |
|---|---|---|---|
| Experiment 1: simultaneous Imaging of neuronal Ca dependent fluorescence and hemodynamic changes induced by cocaine (Group-Neuron: G(N) Wild-type mice) | NA | a. Saline solution (0.1 ml, i.v.) ($n = 3$) | [HbO$_2$], [HbR], [HbT] changes |
| | AAV5.Syn.GCaMP6f.WPRE.SV40 | b. Cocaine hydrochloride (1 mg/kg, i.v.) ($n = 5$) | Ca$^{2+}$-G(N) fluorescence; [HbO$_2$], [HbR], [HbT] Vascular diameter changes |
| Experiment 2: simultaneous Imaging of astrocytic Ca dependent fluorescence and hemodynamic changes induced by cocaine (Group-Astrocyte: G(A) GFAP-Cre mice) | NA | a. Saline solution (0.1 ml, i.v.) ($n = 3$) | [HbO$_2$], [HbR], [HbT] changes |
| | AAV5.CAG.Flex.GCaMP6f.WPRE.SV40 | b. Cocaine hydrochloride (1 mg/kg, i.v.) ($n = 5$) | Ca$^{2+}$-G(A) fluorescence; [HbO$_2$], [HbR], [HbT] Vascular diameter changes |
| Experiment 3: Simultaneous Imaging of astrocytic Ca dependentfluorescence, hemodynamic and vascular response to cocaine with and without GFAP- DREADD (Gi) active by clozapine pretreatment (Group- DREADD: G(D) GFAP-Cre mice) | AAV5.CAG.Flex.GCaMP6f.WPRE.SV40 AAV5.GFAP. hM4D(Gi). mCherry | a. First: pretreatment with vehicle followed by Cocaine (1 mg/kg, i.v.); b. Second: 2 hours later pretreatment with clozapine (0.1 mg/kg; 0.16 ml, i.p.) followed by cocaine (1 mg/kg, i.v.) ($n = 5$) | Ca$^{2+}$-G(D) fluorescence; [HbO$_2$], [HbR], [HbT] Vascular diameter changes |
| Experiment 4: Simultaneous Imaging of neuronal Ca dependent fluorescence and vascular diameter changes induced by clozapine (Group-Clozapine: G(C) WT mice) | AAV5.Syn.GCaMP6f.WPRE.SV40 | Clozapine (0.1 mg/kg; 0.16 ml, i.p., $n = 3$) | Ca$^{2+}$-G(C) fluorescence Vascular diameter changes |

as a function of time in response to clozapine. One-way repeated ANOVA showed no significant differences on vessel size after clozapine injection. Quantification analysis of average efficiency Fig. S4d shows that there was no significant difference between before ($0.2778 \pm 0.4723\%$) and after ($2.829 \pm 1.458\%$) clozapine injection ($p = 0.171$).

## Discussion

Here we show that acute cocaine triggered arterial and venous vasoconstriction and reduced tissue oxygenation. Whereas the later recovered within thirty minutes of injection followed by an overshoot, the vasoconstriction persisted throughout the measurement period. Simultaneous measures of the neuronal or the astrocytic Ca$^{2+}$ dependent fluorescence responses to cocaine revealed a temporal association between the changes in tissue oxygenation and neuronal activation and between the persistent vasoconstriction and the long-lasting astrocytic activation. Specifically, the increased neuronal activation triggered by cocaine was associated with a parallel reduction in tissue oxygenation and the subsequent neuronal inhibition with increased tissue oxygenation despite persistent vasoconstriction, which was associated with astrocytic activation. Moreover, astrocytic inhibition with clozapine in GFAP-DREADD (Gi) expressed mice prevented cocaine-induced vasoconstriction, indicating that astrocytic activation mediated the vascular changes in response to cocaine.

Cocaine induced vasoconstriction reduces cerebral blood flow[37] and can lead to neurological complications such as ischemia and stroke[31,38,39]. Here, we also documented a significant decrease in cerebral blood volume after acute cocaine that persisted 60 min after its administration. Though we also showed

a parallel initial decrease in tissue oxygenation, the hypoxemia recovered by ~25 min followed by an overshoot over its baseline despite the persistent vasoconstriction. The divergence in their temporal sequence indicates that the coupling between tissue oxygenation and blood volume was disrupted by cocaine - an effect that, we postulate, reflects distinct effects of cocaine in neurons and in astrocytes. Cocaine-induced reduction in cerebral blood flow/volume in parallel with neuronal activation could render neuronal tissue particularly vulnerable to hypoxemia, facilitating neurotoxicity with repeated administration[40,41].

Although cocaine's vasoconstricting effects are well documented[2,12,42], the cellular mechanisms that underlie it remain elusive. Accumulating evidence indicates that drug exposure can have dynamic and long-lasting effects on astrocytes and other glial cells[43]. However, the mechanism underlying the effects of cocaine on vasoconstriction as well as those underlying astrocytic $\Delta F/F$ [Ca$^{2+}$]$_i$ increases are still not understood. Cocaine's vasoconstriction effects are likely to reflect its sympathomimetic effects though its effects on L-type Ca channels in blood vessels are also likely to contribute[44]. Astrocytic $\Delta F/F$ [Ca$^{2+}$]$_i$ accumulation associated with dopamine signaling involves Ca$^{2+}$ release from internal stores via the Gq-PLC pathway[45,46]. Ionotropic receptors and voltage-gated Ca$^{2+}$ channels also mediate Ca$^{2+}$ influx into astrocytes[47] and have been implicated in cocaine's effects[48]. Recently, we showed[49] that Ca$^{2+}$ channel blockade reduced cocaine's vasoconstriction and neurotoxicity in the prefrontal cortex. Others have also reported that Ca$^{2+}$-channel blockers reduce negative outcomes from cocaine-induced cerebral ischemia and stroke by buffering cocaine-induced vasoconstriction[50], thus indicating involvement of ionic homeostasis in cocaine-induced vasoconstriction.

Additionally, cocaine triggers neuroadaptations[48,51] in glutamate neurotransmission that is regulated by astrocytes that could further worsen cocaine induced neurotoxicity. These complex roles of astrocytes in brain under cocaine-induced pathophysiological condition require further investigation. Here, we show that astrocytic activation by cocaine underlies cocaine-induced vasoconstriction as evidenced by the temporal correspondence between astrocytic activation and the reduction in arterial blood volumes and the prevention of cocaine induced vasoconstriction with inhibition of astrocytic activation. These findings are clinically relevant for they suggest that interventions to reduce astrocyte activation by cocaine might help restore cerebral blood flow in cocaine users.

Neuronal responses to cocaine differed from those of astrocytes and had a distinct association with vascular changes. Specifically, cocaine initially triggered neuronal activation that lasted approximately 30 min and was associated with reduced tissue oxygenation subsequently followed by neuronal inhibition and increased tissue oxygenation. This indicates that the changes in oxygen metabolism due to cocaine effects on neuronal activation and inhibition underlie the changes in tissue oxygenation. The duration of neuronal activation that we observed with acute cocaine is consistent to the pharmacokinetics of cocaine in brain when given intravenously[52,53] and to the duration of striatal dopamine increases[3,31] and of locomotor activation. Neuronal activation with acute cocaine is also consistent with studies that used manganese-enhanced MRI[54]. The neuronal inhibition that follows the initial activation might underlie the reductions in brain glucose metabolism reported during cocaine withdrawal[55,56].

The role of astrocytes in brain function including in neurovascular coupling is increasingly recognized[57]. Astrocytes contribute to cerebrovascular vasodilation during neurovascular coupling and to the vasoconstriction that subsequently restores vascular tone[35,58]. However studies on the effects of cocaine in astrocytes have mostly focused on synaptic and circuitry regulation associated with addiction-related behaviors[59]. Moreover, very few studies have investigated the effects of acute cocaine in astrocyte activity and to our knowledge no in vivo study has been published. An in vitro study done in slices from the nucleus accumbens incubated with cocaine reported increases in $Ca^{2+}$ transients in astrocytes[60]. Though these results are consistent with our in vivo findings, comparisons are constrained by the fast pharmacokinetics of cocaine in vivo versus stable cocaine levels in in vitro preparations. Nonetheless both document the sensitivity of astrocytic $Ca^{2+}$ transients to cocaine administration.

Our study showed that the astrocytic $Ca^{2+}$ dependent fluorescent changes in response to cocaine (i.e., maximal $\Delta F/F_{Ca^{2+}-G(A)} = 2.106 \pm 0.4333\%$, $n = 5$) were lower than the neuronal $Ca^{2+}$ fluorescent changes (maximal $\Delta F/F_{Ca^{2+}-G(N)} = 6.065 \pm 1.463\%$, $n = 5$, Fig. 4). To minimize expression difference between these two groups of animals, we delivered an identical viral volume into the cortex of all animals. Our ex vivo experiments (Supplementary Fig. 1) indicated that there were no significant differences of GCaMP6f expression into neurons and astrocytes ($n = 3$ animals/per group, ROIs=5/animal, $p = 0.94$). In addition, the $Ca^{2+}$ dependent fluorescent time course was quantified as percent change relative to baseline to eliminate potential baseline variations between animals. Taken together, the amplitudes of fluorescent changes in G(N) and G(A) animals in response to cocaine injection should represent the cocaine-induced intracellular calcium changes in neuron and astrocytes, respectively. Indeed, we previously reported that cortical $Ca^{2+}_N$ and $Ca^{2+}_A$ fluorescent responses to sensory stimulation were stronger for neurons ($\Delta F/F_N = 6.4 \pm 0.29\%$) than astrocytes ($\Delta F/F_A = 1.7 \pm 0.1\%$), supporting difference in cellular Ca responses between these two cell types[35].

One limitation of our study was that measurements were done in anesthetized mice using isoflurane. Thus, vasodilation from isoflurane might have accentuated the magnitude of cocaine induced vasoconstriction and its anesthetic effects might have attenuated the sensitivity of neurons and perhaps also of astrocytes to cocaine. Thus future studies should evaluate cocaine's effects in awake animals. Another limitation was that we imaged neurons and astrocytes with GCaMP6f in separate group of animals, which was done to optimize detection of $Ca^{2+}$ fluorescence changes as a function of cell type. In other words, it allowed us to avoid using the same gain setting fitted to the stronger fluorescence from one cell type (e.g., from neurons) relative to the weaker ones from another cell type (e.g., from astrocytes). Nevertheless, red-shifted genetically encoded calcium indicator such as jRGECO1a red fluorescence probe could be used with green probes such as GCaMP6f to image the activity and interactions of different cell types such as neurons and astrocytes simultaneously. To do so, custom-designed emission filters will be needed to synchronize with the excitation of GCaMP and jRGECO1a, respectively, so that it can detect images of green $Ca^{2+}$ fluorescence from astrocytes and red $Ca^{2+}$ fluorescence signal from neurons in the same field of view. Another limitation was the small sample size used in this study driven by laboriousness of the imaging experiments. Nevertheless, we were able to show significant effects of cocaine, which reflects the large effects being measured.

In summary, we simultaneously assessed neuronal/astrocytic $Ca^{2+}$ dependent fluorescence, vascular diameter and hemodynamic changes in response to acute cocaine in the cortex of mice in vivo. We show that cocaine evoked neuronal activation and subsequent deactivation were accompanied by parallel decreases and increases in tissue oxygenation whereas long-lasting increases in astrocytic $\Delta F/F$ $Ca^{2+}$ activity were associated with persistent vasoconstriction, which was blocked by astrocyte inhibition. This indicates that neurons and astrocytes play different roles in mediating neurovascular coupling in response to cocaine and implicate astrocytes in the long-lasting vasoconstriction associated with cocaine use. Understanding the cellular and vascular interaction induced by cocaine will be helpful for putative treatment strategies to reduce cerebrovascular pathology from cocaine use.

## Methods

**Animals and experimental design.** Transgenic GFAP-Cre mice (Jackson Laboratory, https://www.jax.org/strain/024098) and C57BL/6 wild type (WT) mice (Jackson Laboratory) were used for this study when they reached ~8 weeks of age. Experimental mice were divided into four groups as shown in Table 1. All experimental procedures were approved by the Institutional Animal Care and Use Committee at Stony Brook University and conducted according to the National Institutes of Health (NIH) Guidelines for Care and Use of Laboratory Animals.

**Viral injection into mouse's cortex.** To detect neuron- and astrocyte-specific $[Ca^{2+}]_i$ fluorescence response to acute cocaine challenge, we used the genetically encoded calcium indicator, GCaMP6f via viral delivery into mouse cortex. Specifically, 0.5 μl virus, AAV5.CAG.Flex.GCaMP6f.WPRE.SV40 (AV-5-PV2816, Penn Vector Core, Fig. 1c), was slowly infused at 0.2 μl/min into transgenic mice (GFAP-Cre, Group-A(b) in Table 1) to express GCaMP6f in astrocytes through a small hole predrilled in the skull overlaying the right somatosensory cortex [A/P: + 1.7; M/L: −1.5; D/V: −0.5 mm]; 0.5 μl virus, AAV5.Syn.GCaMP6f.WPRE.SV40 (AV-5-PV2822, Penn Vector Core), was infused into the same brain region of wild type mice (C57BL/6, Group-N(b) in Table 1) to express GCaMP6f into neurons. To examine whether astrocytic $Ca^{2+}$ was involved in cocaine-induced vasoconstriction, a mixture of two viruses (AAV5.CAG.Flex.GCaMP6f.WPRE.SV40 with AAV5.GFAP. hM4D (Gi).mCherry 0.5 μl /each virus) was injected into the same region of cortex (GFAP-Cre, Group-D in Table 1). To assess clozapine's effects on neuronal Ca dependent fluorescence signaling and vascular diameter changes, AAV5.Syn.GCaMP6f.WPRE.SV40 was infused into the cortex of wild type mice (C57BL/6, Group-C in Table 1) to express GCaMP6f into neurons. During viral injection, mice were anesthetized with inhalation of 2% isoflurane mixed with pure

oxygen and their heads mounted on a stereotaxic frame while we monitored their physiology. After completion of procedure the mice were monitored daily for a few days to ensure they were healthy.

**Animal preparation for optical imaging**. After 4–6 weeks of viral injection, a cranial optical window (~2 × 2 mm²) was created on the skull over the viral injection spot to allow for optical imaging[61]. Surgical procedures were performed under isoflurane anesthesia. The mouse head was placed on a stereotactic frame and the skull surrounding the viral injection spot was carefully thinned with a dental drill and then removed to expose the brain. Saline was applied to the brain tissue and the open cortex was immediately covered with a sterile coverslip (4 × 3 mm²) sealed with biocompatible glue. During surgery, the animal's physiology was continuously monitored, including respiration and body temperature to ensure they were physiologically stable. After window implantation, a catheter was placed on a tail vein for drug administration. Then the animal was positioned on a custom head mount for in vivo image acquisition.

**Simultaneous imaging of neuronal/astrocytic [Ca²⁺] fluorescence and cortical hemodynamic changes in response to cocaine**. A multimodality imaging platform developed in our laboratory[36,37,62,63] was used to simultaneously image astrocytic $\Delta F/F_{(\Delta Ca^{2+}-G(A))}$ or neuronal $\Delta F/F_{(\Delta Ca^{2+}-G(N))}$ fluorescence, oxygenated hemoglobin [HbO₂], and deoxygenated hemoglobin [HbR] from the mouse cortex. As illustrated in Fig. 1a, the light beams at three wavelengths of 488 nm, 568 nm, and 630 nm were delivered to the cortex separately with a time-sharing mode (10 ms exposure time per channel). The GCaMP6f fluorescence was excited at $\lambda_{ex} = 488$ nm with the emission peaked within 512–535 nm along with the reflectance (from $\lambda_1 = 568$ nm, and $\lambda_2 = 630$ nm) were detected by the sCMOS camera (synchronized with the illumination paradigm), respectively. The reflectance at $\lambda_1 = 568$ nm and $\lambda_2 = 630$ nm were used to extract changes in [HbO₂] and [HbR] (Fig. 2a) in mouse cortex[36,64]. Also, the arteries and veins can be separately distinguished from the images obtained at $\lambda_1 = 568$ nm and $\lambda_2 = 630$ nm (Supplementary Fig. 5). To assess cocaine-induced dynamic changes in astrocytic or neuronal Ca²⁺ fluorescence and brain hemodynamics, we administered cocaine (1 mg/kg, i.v.) through the tail vein followed by 0.5 ml saline to ensure delivery of the cocaine in the catheter. A total of 70 min duration of images were acquired for each experiment, including 10 min baseline before cocaine and 60 min post cocaine administration.

To investigate astrocyte's involvement in cocaine-induced vasoconstriction, animals in Experiment 3 (Table 1) underwent two sets of imaging session with at least 110 min between each cocaine infusion (a: VEH (saline) pretreatment followed by cocaine challenge; b: clozapine pretreatment (0.1 mg/kg;0.16 ml, i.p. 30 min) followed by cocaine challenge. During the experiment, astrocytic Ca²⁺ dependent fluorescence ($\Delta F/F_{[Ca^{2+}]-G(D)}$), vessels' diameter and cortical hemodynamic changes at $\lambda_1 = 568$ nm, and $\lambda_2 = 630$ nm were detected before and after cocaine challenges. In addition, to access the effects of clozapine on neuronal Ca²⁺ fluorescence and vascular diameters, the neuronal Ca²⁺ dependent fluorescence ($\Delta F/F_{[Ca^{2+}]-G(C)}$) and cortical hemodynamic changes were recorded at baseline (10 mins) and after clozapine (30 mins) in Experiment 4 ($n = 3$).

**Image and data processing**. For each experiment, five data sets were obtained from each mouse cortex: (1) [Ca²⁺] dependent fluorescence intensity change from astrocytes ($\Delta F/F_{[Ca^{2+}-G(A)]}$) or neurons ($\Delta F/F_{[Ca^{2+}-G(N)]}$), which reflects astrocytic or neural activation, (2) changes in vessel diameter; (3) changes in total blood volume, ΔHbT within arteries and veins, (4) changes in oxygenated-hemoglobin ΔHbO₂, and (5) deoxygenated hemoglobin ΔHbR to assess tissue oxygenation status in response to cocaine (1 mg/kg, i.v.). Data from different animals in each group were averaged and the amplitude of each signal and duration of responses to cocaine were quantified. We also compared these parameters between the astrocytic and neuronal expressed animal groups.

All multi-channel (i.e., multi-wavelength) images were acquired using the time-sharing strategy by multimodal imaging platform which were then regrouped to time-lapse image sets for each channel. For GCaMP6f-Ca²⁺ fluorescence imaging, five regions of interest (ROIs, pink dots as illustrated in Fig. 4a, b) were selected within the GCaMP6f-expressing cortical areas devoid of visible blood vessels to track the temporal changes of astrocytic or neuronal fluorescence induced by cocaine. To control for absorption differences due to hemodynamic changes, such as in HbT, values in regions expressing GCaMP6f were divided by the region of the cortex, where there was no expression (white triangle illustrated in Fig. 4a, b). The neuronal or astrocytic Ca fluorescent activity was quantified as $\Delta F/F$ ($\Delta F/F_{[Ca^{2+}-G(N)]}$, $\Delta F/F_{[Ca^{2+}-G(A)]}$, respectively). The relative changes of fluorescence signal over its baseline (before cocaine), $\Delta F/F(\Delta[Ca^{2+}])=[F(t)-F_{baseline})/F_{baseline}) \times 100]\%$were calculated for each animal to eliminate effects of variations in GCaMP6f expression on fluorescence intensity changes in response to cocaine.

To access cocaine-induced vasoconstriction and examine whether astrocytic Ca inhibition using GFAP-DREADD (Gi) (activated by clozapine before cocaine administration) would reduce the vasoconstriction effects of cocaine on cerebral vessels, the vessel diameters were quantified using a custom Matlab program. Specifically, multiple ROIs were selected across the blood vessels in different vessel

types (e.g., veins and arteries to track vessel diameters starting from 10 min baseline period to 60 min post cocaine). The vessel diameter was estimated using images obtained from $\lambda_1 = 568$ nm with a Gaussian process approach. The detail algorithm of computation has been published by Asl et al.,[65] Briefly, it places $x$ and $y$ coordinates of a seed point at the center of a vessel at the selected location of the ROI. The centerline and direction of the vessel are then computed using the Radon transform, and train MATLAB to start tracking from the input seed point to the next 10 points ($\phi_i$, $i = 1$–10, within ~60 μm) lying in the center of the blood vessel. The diameter $\phi_i$ at each point is then estimated using a separate Gaussian process. The mean volume of $\phi_i$ ($i = 1$–10) was calculated as the vessel diameters at the location selected ROIs. To track time course of vessel diameter changes, a while-loop was added to read from the first image to the last one (e.g., $m = 4200$ frames or based on our imaging time for each experiment above). As a result, there were 10*m data points computed to present the blood vessel change along the time course of imaging, which represents the vessel diameter changes as a function of time at the location of selected ROI. In other words, the vessel diameter at each ROI was continually tracked in each frame and its change was quantified over its baseline (before cocaine) as a function of time following cocaine injection.

The hemodynamic responses of ΔHbO₂ and ΔHbR were calculated from the acquired $\lambda_1 = 568$ nm and $\lambda_2 = 630$ nm images. To minimize the artifacts induced by the hemodynamic changes due to cocaine infusion, vehicle animals were used in Experiments1–2 in Table 1. The mean hemodynamic changes in response to saline (0.1 ml, i.v., Experiments 1a, 2a) were used to correct the temporal absorbance changes due to cocaine injection (e.g., Experiments 1b, 2b). Supplementary Figure 6 illustrates the correction procedures for one of the animals. The correction was conducted in the raw HbO₂ (568 nm) and HbR (630 nm) channels for each animal. Meanwhile, Supplementary Fig. 7 shows the comparison of time courses of ΔHbT before correction (Supplementary Fig. 7a, c) and after correction (Supplementary Fig. 7b, d), it indicates the infusion correction slightly reduces ΔHbT changes within injection period ($t < 2$ mins). After correction, ΔHbO₂ and ΔHbR were calculated based on the following equation:

$$\begin{bmatrix} \Delta[HbO_2(t)] \\ \Delta[HbR(t)] \end{bmatrix} = \begin{bmatrix} \varepsilon_{HbO_2}^{\lambda_1} & \varepsilon_{HbR}^{\lambda_1} \\ \varepsilon_{HbO_2}^{\lambda_2} & \varepsilon_{HbR}^{\lambda_2} \end{bmatrix}^{-1} \cdot \begin{bmatrix} \frac{\ln(R_{\lambda_1}(0)/R_{\lambda_1}(t))}{L_{\lambda_1}(t)} \\ \frac{\ln(R_{\lambda_2}(0)/R_{\lambda_2}(t))}{L_{\lambda_2}(t)} \end{bmatrix} \quad (1)$$

where $\varepsilon^{\lambda}_{HbO2}$, $\varepsilon^{\lambda}_{HbR}$ are the molar extinction coefficients for HbO₂ and HbR, and $R_{\lambda1}(t)$, $R_{\lambda2}(t)$ are the measured diffuse reflectances at these wavelengths. $R_{\lambda1}(0)$, $R_{\lambda2}(0)$ are their baseline values before cocaine. $L_{\lambda1}(t)$, $L_{\lambda2}(t)$ are their pathlengths[66,67]. The total hemoglobin concentration change can be obtained by

$$\Delta HbT = \Delta HbO_2 + \Delta HbR \quad (2)$$

which is also called the cerebral blood volume change within the cortex[62].

Nine ROIs were selected from arteries, veins and tissue (three from each component as illustrated in Fig. 2b) to capture the hemodynamic responses to cocaine (ΔHbO₂, ΔHbR, ΔHbT) from vascular and tissue compartments.

**Immunohistochemistry and Ex-vivo Imaging**. After in vivo imaging, the mouse was transcardially perfused with 0.1 M phosphate-buffered saline (PBS), then fixated with 4% paraformaldehyde (PFA) in 0.1 M PBS overnight. The cryoprotected brain was immersed in 30% sucrose solution and sectioned to 50 μm thick slices. For immunostaining, sectioned brain tissues were treated with the primary chicken anti-GFP antibody (1:200, Thermofischer) to enhance GCaMP6f fluorescence followed by specific secondary antibody 488 anti-chicken (1:200, Jackson Immunoresearch). After that, for GCaMP6f expressed in astrocytes brain slices, rabbit anti-GFAP (1:200, Millpore) was used meanwhile for GCaMP6f expressed in neurons samples, mouse anti-Neun (1:200, Millpore) was used. Both primary antibodies followed by their respective secondary antibodies (i.e., 594 anti-rabbit or 594 anti-mouse to visualize the location of astrocytes or neurons). Ex vivo Images were acquired with a confocal microscope.

**Statistics and reproducibility**. All data are presented as means ± SEM. Comparisons between two groups (e.g., astrocytic GCaMP6f-expressed group -Group A vs. neuronal GCaMP6f-expressed group -Group N) were analyzed using an unpaired t-test. Comparisons between multiple groups (>2 groups) were analyzed using One Way ANOVA. A $P$-value <0.05 was considered statistically significant for all cases. For cross-correlation between two temporal traces (e.g., $(\Delta F/F_{[Ca^{2+}-G(N)]}$ vs ΔHbO₂-G(N)), the Pearson correlation was calculated. A $P$-value <0.05 was considered significant. Sample size in each experiment are indicated in each figure legend and summarized in Table 1.

**Reporting summary**. Further information on research design is available in the Nature Research Reporting Summary linked to this article.

## Data availability
All data generated or analyzed during this study are included in this published article (and its supplementary information files), which can be also available from the

corresponding author on reasonable request. Source data underlying plots shown in Figs. 2–7 are provided in Supplementary Data 1.

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

## Acknowledgements

This work was supported in part by National Institutes of Health (NIH) grants 2R01 DA029718 (C.D. and Y.P.), RF1DA048808 (Y.P. and C.D.), R21DA042597 (C.D. and Y.P.) and NIH's Intramural Program of NIAAA (N.D.V.). The authors would like to thank Y.Q. Yan for partially assisting KP on data process and NIDA's Drug Supply Program for providing cocaine used in this study.

## Author contributions

C.D., N.D.V., and Y.P. designed the research; Y.L., Y.H., and K.P. carried out the in vivo experiments and conducted image processing and analysis. Y.L., C.D. N.V.D., and Y.P. contributed to data interpretation, result discussions, and manuscript writing.

## Competing interests

The authors declare no competing interests.
