## [Peer Review File · Communications Biology]

Reviewers' comments:

Reviewer #1 (Remarks to the Author):

The manuscript by Liu et-al used a multi-optical imaging platform and a genetically encoded calcium indicator (GECI) to study the effects of acute cocaine on neuronal and astrocytic activity, tissue oxygenation and vascular hemodynamics in the mouse cerebral cortex. The results showed that cocaine decreases cerebral total blood volume (HbT) and temporally reducing tissue oxygenation. Using cellular imaging authors found that neurons and astrocytes play different roles in mediating neurovascular coupling in response to cocaine and suggested astrocytes in the long-lasting vasoconstriction associated with cocaine use. This is a well designed study and the results will help the understanding of abusive use of drugs. I just have a few comments.

1. The timecourse of vessel diameter change for all experiments should be included in the figures for readers have a better assessment of the relationship of hemodynamic change (HbO, HbT, and HbR) and the structural change (vessel size).
2. It's confusion in Fig. 7, why didn't the author also include the HbO and HbT change together with the vessel diameter change? Did the inhibition of astrocytic ΔCa^{2+} -G(A) by clozapine still induce HbO and HbT change? I believe this result is critical to support the authors claim.
3. How were the arteries and veins determined? For Fig. 2b1, it looks like the identified artery (lower middle pink dot) was indeed a branch of ta vein.
4. The sentence of lines 69-71 is confusion. Please rephrase.
5. The yellow, cyan, and pink dots in Fig. 2b should be explained in the caption.
6. Line 135, "where HbR absorption is higher HbO₂", should be "is higher than HbO₂"

Reviewer #2 (Remarks to the Author):

Cocaine's astrocytic but not neuronal effects are associated cerebrovascular vasoconstriction: In vivo optical imaging of mouse brain
Liu et al.

In their manuscript, the authors describe the effect of acute cocaine administration on total blood volume, neuronal and glial calcium concentration in the cerebral cortex in vivo. Cocaine induced a large, transient calcium increase in neurons and a smaller, long-lasting calcium increase in astrocytes. These calcium signals were accompanied by a decrease in total blood volume and tissue oxygenation. The time course of cocaine-evoked decrease in tissue oxygenation follows the time course of neuronal calcium signals while the time course of total blood volume in arteries follows the time course of calcium signaling in astrocytes. Cocaine fails to evoke vasodilation after expression and activation of Gi-DREADD in astrocytes. The authors conclude that cocaine-evoked activation of astrocytes leads to vasoconstriction.

The manuscript contains some interesting aspects but I have several concerns that should be addressed by the authors.

Methods

- 1) GcaMP6f fluorescence is collected using a 512-535 nm emission filter which should block the 568 and 630 nm reflected light of the HbO₂ and HbR images. Did the authors use different filters (filter wheel) for the three image sequences (488, 568, 630 nm)? Please add details of the components of the emission path of the optical setup.
- 2) Give more details how the vessel size was detected in the last set of experiments (Fig. 7). The authors state that ROIs were selected IN the blood vessels. How can the diameter be measured if the ROI is in the vessel and does not span the entire range of the blood vessel? In addition, does the Matlab routine automatically detect the border of the blood vessel?
- 3) According to the description in the methods the authors do not represent calcium changes in percent (%) as stated, but simply by $\Delta F/F$ which lacks any unit. One can give these figures in % by multiplying them with 100, i.e. $\Delta F/F$ of 3 reflects a change in fluorescence of 300 %. This should be changed throughout the manuscript and in the figures (labeling of axes).

Results and conclusions

4) Fig. 5b shows a clear correlation between neuronal calcium and tissue oxygenation. The authors conclude that „cocaine-induced reduction in tissue oxygen content in the peak response period was due to increase in neuronal activation“ (p. 9, l. 196). A correlation does not indicate a causal connection. Indeed, the causal connection could be vice versa, with an increase in neuronal activation due to a reduction in tissue oxygenation and hence metabolic stress. The authors should change the manuscript accordingly to avoid any premature conclusions.

5) Fig. 6b. The same applies for the correlation between astrocytic calcium and total blood volume in arteries. The authors state that „astrocytic activation contributed to cocaine-induced vasoconstriction“ (p. 10, l. 213). Again, the correlation does not prove a causal connection.

6) The authors used calcium imaging to visualize astrocyte activation. The mechanism of cocaine-evoked calcium signaling in astrocytes is not clear, but it is likely that it involves calcium release from internal stores via the Gq-PLC pathway. One action of cocaine is inhibition of dopamine uptake, and the accumulated extracellular dopamine could activate astrocytic dopamine receptors and trigger Gq-coupled calcium release (Jennings et al. *Glia*. 2017 Mar;65(3):447-459. doi: 10.1002/glia.23103; Fischer et al. *Sci Rep*. 2020 Jan 20;10(1):631. doi: 10.1038/s41598-020-57462-4). Therefore, it is not clear why Gi-DREADD would inhibit astrocyte calcium signaling since Gi targets adenylate cyclase, not PLC. While it is shown that „inhibitory“ Gi-DREADD reduce excitation of neurons, the effect of Gi-DREADD and hence adenylate cyclase inhibition in astrocytes is barely studied and it can not be considered as evidenced that Gi-DREADD indeed inhibit astrocyte activation. I recommend that the authors perform control experiments to show whether Gi-DREADD activation suppresses cocaine-evoked calcium signaling in astrocytes.

7) Fig. 7. Why did the authors change the read-out of vasoconstriction in their last set of experiments (measurement of blood vessel diameter instead of deltaHbT)? There are no control experiments showing the response to cocaine without Gi-DREADD for the change in vessel diameter. Please provide those controls. Furthermore, what is the effect of Gi-DREADD on deltaHbT in arteries? Without those experiments the results are not conclusive and the study rather preliminary.

Terminology

8) The authors use the term „Ca²⁺ fluorescence“ throughout the manuscript, they state that (GCaMP6f) fluorescence „was from astrocytic Ca²⁺“ and describe the anti-GFP immunostaining of GCaMP6f in neurons and astrocytes, respectively, as „neuronal Ca²⁺“ and „astrocytic Ca²⁺“ (Fig. 1), and so on. Ca²⁺ is non-fluorescent, and any fluorescence the authors refer to derives from GCaMP6f, not Ca²⁺ itself. Hence, the authors should go through the manuscript and change the terminology to make clear what they are talking about. However, Ca²⁺-dependent changes in fluorescence can be considered as calcium signals. Thus, it makes a difference whether one refers to fluorescence or to Ca²⁺-dependent fluorescence changes.

Supplements

9) Fig. S1: The correct lettering of the figure panels following f is g, h, i (not j, ...).

10) Fig. S1 (j-I): „... the arrowheads (show) the connection between the astrocytes ...“ There are no arrowheads in the figure.

11) Fig. S1 (j-I): The legend states that the depicted images are magnifications from the images in d-f. I don't see structures in d-f that resemble those in j-I. Please indicate the region in d-f that is magnified in j-I, e.g. by a square as overlay.

Title

12) Place "with" after "associated".

Reviewer #3 (Remarks to the Author):

In this manuscript, the authors used a multimodal imaging platform to simultaneously measure neuronal and astrocytic Ca²⁺ signals, tissue oxygenation and vascular hemodynamics in response to acute application of cocaine. The authors reported that while cocaine reduced cerebral total blood volume, it only temporally decreased tissue oxygenation that became overshoot after

returned to baseline. The overshoot was observed despite a sustained drop in total blood volume. Interestingly, while cocaine caused a biphasic neuronal Ca²⁺ response, cocaine did not elicit the same biphasic Ca²⁺ response in astrocytes. It was also reported that inhibition of astrocytic activity with GFAP-DREADD (Gi) blocked cocaine-induced vasoconstriction. This is an interesting study with novel findings that provide insights into the interactions between astrocytes and the vasculature in response to cocaine and perhaps NVC in general. Below are my comments.

1. The basal neuronal Ca²⁺ signals appeared to be much stronger than that of astrocytic Ca²⁺ signals, would the authors care to elaborate on the difference a bit? Would this affect the differences seen in cocaine-induced responses between the 2 cell types?
2. It would be more informative if the authors could show the astrocytic Ca²⁺ signals in the GFAP-DREADD experiments.
3. Did application of Clozapine affect the vessel diameter?
4. Did application of Clozapine affect neuronal Ca²⁺?

Minor:

It should be DREADD and not DRADD.

Title: " Cocaine's astrocytic but not neuronal effects are associated **with** cerebrovascular vasoconstriction: In vivo optical imaging of mouse brain"

Tracking #: COMMSBIO-22-0447-T

Authors: Yanzuo Liu, Yueming Hua, Kicheon Park, Nora D. Volkow, Yingtian Pan, Congwu Du

Response to the Referee Comments

We are very pleased that the reviewers recognized the unique aspects of this work. We also thank the reviewers for their valuable comments and suggestions to our paper, which we have addressed in this revised version. We have carefully responded to each reviewer's questions and comments, some of which required that we do extra experiments. The results from these new experiments are either included as part of the main results section or as part of the supplemental material and referred to in the discussion section. Revisions are highlighted in red in the revised manuscript. Below are our point-to-point responses to the referee comments.

Reviewers' comments:

Reviewer #1 (Remarks to the Author):

*The manuscript by Liu et-al used a multi-optical imaging platform and a genetically encoded calcium indicator (GECI) to study the effects of acute cocaine on neuronal and astrocytic activity, tissue oxygenation and vascular hemodynamics in the mouse cerebral cortex. The results showed that cocaine decreases cerebral total blood volume (HbT) and temporally reducing tissue oxygenation. Using cellular imaging authors found that neurons and astrocytes play different roles in mediating neurovascular coupling in response to cocaine and suggested astrocytes in the long-lasting vasoconstriction associated with cocaine use. **This is a well-designed study and the results will help the understanding of abusive use of drugs.** I just have a few comments.*

1. The time course of vessel diameter change for all experiments should be included in the figures for readers have a better assessment of the relationship of hemodynamic change (HbO, HbT, and HbR) and the structural change (vessel size).

Thanks for the suggestion. These are specific changes done for this purpose.

Diameter changes are direct measurements of vessel morphology, whereas hemodynamic changes, e.g., changes in HbT (ΔHbT) are measures of changes in total absorbance (e.g., total blood volume changes) within vessels, and unlike diameter detection, it is sensitive to dynamic changes of absorbers, for example, ΔHbT is decreased temporally during saline injection to result in the temporal total blood volume changes as shown in (Fig.r1).

To compare cocaine-induced vessel diameter changes with hemodynamic responses, the following strategies were applied:

1) Correcting the effects of blood volume changes due to injection: To mimic the disruption of cocaine's infusion to blood volume in the cortex, saline solution with the same volume as for the cocaine injection (i.e., 0.1ml) was infused into the vehicle group animals in Experiments 1-2(a) (see updated **Table 1** below).

Table 1: Animal groups and experimental design

Experiment	Pre-Virus Injection	Drug Challenge	Detection/Imaging
Expt1: Simultaneous Imaging of neuronal Ca dependent fluorescence and hemodynamic changes induced by cocaine (Group-Neuron: G(N) WT mice)	NA	a. Saline solution (0.1ml, i.v.) (n=3)	[HbO₂], [HbR], [HbT] changes
	AAV5.Syn.GCaMP6f.WPRE.SV40	b. Cocaine hydrochloride (1mg/kg, i.v.) (n=5)	Ca ²⁺ -G(N) fluorescence; [HbO ₂], [HbR], [HbT] Vascular diameter changes
Expt2: Simultaneous Imaging of astrocytic Ca dependent fluorescence and hemodynamic changes induced by cocaine (Group-Astrocyte: G(A) GFAP-Cre mice)	NA	a. Saline solution (0.1ml, i.v.) (n=3)	[HbO₂], [HbR], [HbT] changes
	AAV5.CAG.Flex.GCaMP6f.WPRE.SV40	b. Cocaine hydrochloride (1mg/kg, i.v.) (n=5)	Ca ²⁺ -G(A) fluorescence; [HbO ₂], [HbR], [HbT] Vascular diameter changes
Expt3: Simultaneous Imaging of astrocytic Ca dependent fluorescence, hemodynamic and vascular response to cocaine with and without GFAP- DREADD (Gi) active by clozapine pretreatment (Group-DREADD: G(D) GFAP-Cre mice)	AAV5.CAG.Flex.GCaMP6f.WPRE.SV40 AAV5.GFAP. hM3D(Gi). mCherry	a. First: pretreatment with vehicle followed by Cocaine (1mg/kg, i.v.); b. Second: 2 hours later pretreatment with clozapine (0.1mg/kg; 0.16ml, i.p.) followed by cocaine (1mg/kg, i.v.) (n=4)	Ca ²⁺ -G(D) fluorescence; [HbO ₂], [HbR], [HbT] Vascular diameter changes
Expt4: Simultaneous Imaging of neuronal Ca dependent fluorescence and vascular size changes induced by clozapine Group-Clozapine: G(C) WT mice)	AAV5.Syn.GCaMP6f.WPRE.SV40	Clozapine (0.1mg/kg;0.16ml, i.p.) (n=3)	Ca ²⁺ -G(C) fluorescence Vascular diameter changes

Figure r1. Demonstration of correcting blood volume changes due to injection. a) Reflectance ($\ln(R_{\lambda}(0))/R_{\lambda}(t)$) at wavelength of 568nm in response to cocaine (yellow shadow) or saline (green shadow), respectively. a') Cocaine-induced reflection changes after correction at wavelength of 568nm. b) Reflectance ($\ln(R_{\lambda}(0))/R_{\lambda}(t)$) at wavelength of 630nm in response to cocaine (purple shadow) or saline (orange shadow), respectively. b') similar correction is conducted for HbR channel (630nm) before calculation on HbO₂ and HbR.

Figure r2. Cocaine-induced Δ HbT changes before and after correction. Time courses of Δ [HbT] in arteries (red) and veins (blue) in response to cocaine (1mg/kg, i.v.) in neuronal (a, a') GCaMP6f-expressed animals (n=5) and astrocytic (b, b') GCaMP6f-expressed animals (n=5),

The mean reflectance changes at wavelength of 568nm and 630 nm in response to saline (e.g., Expt 1a) were used to correct for temporal absorbance changes due to the injection in cocaine animals (e.g., Expt 1b). **Figure r1** illustrates the correction procedures from an animal. The correction was conducted in the raw HBT (568nm) and HbR (630nm) channels for each animal.

Figure r2 shows the comparison of time courses of Δ HbT before correction (Fig. r2 a, b), presented as Fig. 2c and 2d in original submission and after correction (Fig. r2 a' and b'), it indicates the infusion correction slightly reduces Δ HbT changes within injection period ($t < 2$ mins). For example, Δ HbT decreased to $\sim 20\%$ in Fig. r2a' (instead of $\sim 25\%$ before correction, Fig. r2a in G(N) group animals), and decrease to $\sim 18\%$ in Fig. r2b' instead of $\sim 23\%$ before correction, Fig. r2b in G(A) group animals. As the injection-induced blood volume changes predominately within 5 mins after injection (Fig. r1a), it has minimal inference on the recovery time period as shown in Fig. r2. We conducted the correction for all animals, and updated Fig. 2 accordingly. Other analyses related to Δ HbT changes (e.g., in Fig. 5 and Fig. 6) were updated as well.

2) Quantifying the vessel diameters changes for all experiments. The vessel changes in response to cocaine were analyzed for G(N) and G(A) group animals as shown in **Figure r3**, which are

included in Figure 2 of the revised manuscript (Figure 2 e,f).

We included Fig. r1 into the supplemental information (Fig.S5), and updated Fig. 2 by including Fig. r2 and r3 to form a new Fig. 2 in the revised manuscript.

r3)

a)

b)

c)

Figure r3. Quantification of cocaine-induced vessel diameter changes in arteries (red) and veins (blue) in response to cocaine (1mg/kg, i.v.) in neuronal (G(N) group, a) and astrocytic G(A) group, b) GCaMP6f-expressed animals.

2. It's confusion in Fig. 7, why didn't the author also include the HbO and HbT change together with the vessel diameter change? Did the inhibition of astrocytic $\Delta\text{Ca}^{2+}\text{-G(A)}$ by clozapine still induce HbO and HbT change? I believe this result is critical to support the authors claim.

r4

Figure r4. Cocaine induced Vessel size (c, c') changes along with HbO₂ (b, b') and HbT (a, a') before and after astrocytic ΔCa^{2+} inhabitation by clozapine. d, e, f) Comparison of cocaine-induced HbT, HbO₂ and vessel diameter $\Delta\phi$ changes before and after astrocytic ΔCa^{2+} inhabitation by clozapine. Quantification analysis of average efficiency in vessels, hemodynamic response respectively before Gi active (Grey shadow in (a, b, c)) and after Gi active (Pink shadow in (a', b', c')). Yellow shadow indicate significant difference ($p < 0.05$) time period compare to baseline, and green shadow indicate no-significant time period ($p > 0.05$).

To address this question together with Q7 risen from Reviewer #2 below (i.e., 'Please provide those controls'— responses to cocaine before DREADD Gi to be active), required new experiments for experiment #3, Specifically, animals in the new experiment #3 needed two sets of imaging sessions with at least 110 min between each cocaine infusion (**a:** VEH (saline) pretreatment followed by cocaine challenge; **b:** clozapine pretreatment followed by cocaine challenge).

Figure. r4 shows the results from these new experiments #3 (n=4), including cocaine-induced changes in total vessel hemoglobin (ΔHbT), tissue oxygenated-hemoglobin (ΔHbO_2) and vascular size ($\Delta\phi$) as a function of time before and after clozapine (0.1mg/kg; i.p.) to inhibit the astrocytic ΔCa^{2+} (DREADD Gi). For ΔHbT , prior to DREADD Gi activation by clozapine, cocaine immediately reduced ΔHbT to $-18.84\% \pm 1.115\%$, which did not return to baseline over 60 mins after cocaine. However, after 30 mins of clozapine, the cocaine-induced decrease in ΔHbT from t=2 to 8 min ($p < 0.05$, maximum of $-5.514\% \pm 0.5085\%$) and returned to baseline after 8 min ($p > 0.05$). For tissue ΔHbO_2 before clozapine, cocaine reduced ΔHbO_2 to $-11.81\% \pm 1.047\%$ followed by gradual recovery to baseline at 30.4 ± 8.853 min with an overshoot afterwards, which is consisted with previous observation shown in Figure 3 in the manuscript. After clozapine, tissue ΔHbO_2 after cocaine was slightly reduced to $-5.118\% \pm 1.609\%$. Specifically, one-way repeat ANOVA showed that ΔHbO_2 was reduced compared to the baseline within t= 2-7mins, revealing mild and short-lasting effects of cocaine on tissue oxygenation. For Vessel size($\Delta\phi$), before clozapine cocaine decreased it to $-7.557\% \pm 2.463\%$ followed by gradual recovering to baseline at t=60 min, whereas after clozapine cocaine did not change $\Delta\phi$ compared to baseline (Fig. r4 c').

Comparisons of cocaine-induced mean changes in ΔHbT , ΔHbO_2 and $\Delta\phi$ before and after clozapine activation of DREADD Gi are summarized in Fig.r4 d-f, showing that cocaine-induced changes in ΔHbT , ΔHbO_2 and $\Delta\phi$, are significantly reduced after astrocytic ΔCa^{2+} inhibition. Specifically, without or with astrocytic ΔCa^{2+} inhibition, ΔHbT changed from $-9.668 \pm 0.4799\%/min$ to $-0.6927 \pm 0.0933\%/min$ ($P^* \leq 0.001$); ΔHbO_2 from $-6.867 \pm 2.536 \%/min$ to $-2.179 \pm 0.3419 \%/min$, ($P^* = 0.003$) and $\Delta\phi$ from $-3.286 \pm 0.8787\%/min$ to $-0.4476 \pm 0.3379\%/min$ ($P^* = 0.038$);

We have included these new results for experiment #3 to Table 1, and replaced Fig. 7 accordingly in the revised manuscript.

3. **A)** How were the arteries and veins determined? For Fig. 2b1, it looks like the identified artery (lower middle pink dot) was indeed a branch of ta vein.

r5

Figure r5. Demonstration of separating arteries and veins from HbT and HbR images. Red traces: arteries; blue traces: veins.

A): As arteries and veins contain predominately oxygenated-hemoglobin (HbO_2) and deoxygenated-hemoglobin (HbR), respectively, the absorbance difference between HbO_2 and HbR) at different wavelengths are used to separate arteries and veins. Figure 2.a. shows the absorption spectra of HbO_2 (red curve) and HbR (blue curve). At wavelength of 568nm, it is an isosbestic point of HbO_2 and HbR spectra with high absorbance, which is used as HbT channel in our MIP system to detect both arteries and veins (Fig. r5a). In contrast, at wavelength of 630nm, HbR has obviously higher light absorption than HbO_2 , which is more sensitive to vein as so-called 'HbR' channel to distinguish veins (Fig.r5b);). Taking the observation from these two wavelengths allowed us to identify arteries and veins as shown in Fig.r5c (which is presented as Fig. b1).

B): The branch near to the lower middle pink dot (marked as red box in Fig.r5a), was identified as artery because it appears in the 'HBT' channel but not in the 'HbR' channel (Fig.r5b). However, it seems as if they were 'connecting' between an artery and a vein branch, which is likely due to superposing from different depths of these vessel branches from the two dimension projection.

We have included the Fig.r5 in the supplemental figure s6.

4. The sentence of lines 69-71 is confusion. Please rephrase.

The sentence has been rephrased as

Neurovascular coupling (NVC) is involved in modulation of brain function¹. Neuronal-vascular interactions are necessary to maintain an adequate supply of oxygen and glucose for proper neuronal function^{2,3} and until recently most studies on NVC focused on neurons. However, there is now increasing interest to study the interactions between neurons and glial cells and their role in NVC.

5. The yellow, cyan, and pink dots in 2b should be explained in the caption.

Thank you for this suggestion. The Figure 2 caption has been adjusted accordingly. (Pg 7)

“Regions of interest (ROIs) were selected from arteries (e.g., pink dots), veins (e.g., blue dots) and tissue (e.g., yellow dots) from each animal.”

6. Line 135, “where HbR absorption is higher HbO₂”, should be “is higher than HbO₂”

Thanks and we have corrected it, i.e., “where HbR absorption is higher than HbO₂” (in manuscript Pg 5, Ln 140)

Reviewer #2 (Remarks to the Author):

Cocaine’s astrocytic but not neuronal effects are associated cerebrovascular vasoconstriction: In vivo optical imaging of mouse brain
Liu et al.

In their manuscript, the authors describe the effect of acute cocaine administration on total blood volume, neuronal and glial calcium concentration in the cerebral cortex in vivo. Cocaine induced a large, transient calcium increase in neurons and a smaller, long-lasting calcium increase in astrocytes. These calcium signals were accompanied by a decrease in total blood volume and tissue oxygenation. The time course of cocaine-evoked decrease in tissue oxygenation follows the time course of neuronal calcium signals while the time course of total blood volume in arteries follows the time course of calcium signaling in astrocytes. Cocaine fails to evoke vasodilation after expression and activation of Gi-DREADD in astrocytes. The authors conclude that cocaine-evoked activation of astrocytes leads to vasoconstriction.

The manuscript contains some interesting aspects but I have several concerns that should be addressed by the authors.

Methods

1) GcaMP6f fluorescence is collected using a 512-535 nm emission filter which should block the 568 and 630 nm reflected light of the HbO₂ and HbR images. Did the authors use different filters (filter wheel) for the three image sequences (488, 568, 630 nm)? Please add details of the components of the emission path of the optical setup.

Sorry for the confusion, the GCaMP6f emission peaked within 512-535nm, which are not wavelengths used for the emission filter. For emission filter, we used a long-pass (LP) filter with a cutoff at 510nm. The excitation GGaMP6f was at 488 nm, this LP filter can block the excitation light to be detected, and meanwhile it permits the emission of GCaMP6f Ca²⁺ fluorescence as well as the reflectance at 568 and 630nm sequentially by using a time-sharing approach.

We have clarified the following statement in the Method (Pg18 Ln 446-450).

“The light beams at three wavelengths of 488nm, 568 nm, and 630nm were delivered to the cortex separately with a time-sharing mode (10 ms per channel). The GCaMP6f fluorescence was excited at $\lambda_{ex}=488\text{nm}$ with the emission (peaked within 512-535nm) along with the reflectance (from $\lambda_1=568\text{ nm}$, and $\lambda_2=630\text{nm}$) detected by the sCMOS camera (synchronized with the illumination paradigm), respectively.”

Also, we have marked LP filter and added the exposure time per channel in Figure 1, and describe it in the caption in the revised manuscript.

2) **a)** Give more details how the vessel size was detected in the last set of experiments (Fig. 7). **b)** The authors state that ROIs were selected IN the blood vessels. How can the diameter be measured if the ROI is in the vessel and does not span the entire range of the blood vessel? **c)** In addition, does the Matlab routine automatically detect the border of the blood vessel?

a) The vessel diameter is estimated by using the image obtained from $\lambda_1=568\text{nm}$ with a Gaussian process approach. The detail algorithm of computation was published by Asl et al, previously⁴. Briefly, it puts x and y coordinates of a seed point at the center of a vessel as selected location of ROI. The centerline and direction of the vessel are then computed using the Radon transform, and train MATLAB to start tracking from the input seed point to the next 10 points (ϕ_i , $i=1-10$, within $\sim 60\mu\text{m}$) lying in the center of the blood vessel. The diameter ϕ_i at each point is then estimated using a separate Gaussian process. The mean volume of ϕ_i ($i=1-10$) was calculated as the vessel diameters at the location as selected ROIs. To track time course of vessel diameter changes, a while-loop was added to read from the first image to the last one of our images (e.g., $m=4200$ frames or based on our imaging time for each experiment above). As a result, there were $10*m$ data points were computed to present the blood vessel change along the time course of imaging, which represents the vessel size changes as a function of time at the location of selected ROI. The statement has been included in the manuscript now (Pg 19 Ln492-506)

b) ‘Multiple ROIs were selected from different vessel types (e.g., veins and arteries) to track vessel diameters starting from 10min baseline period to 60min post cocaine’, which we clarify in the revised manuscript (Pg 19, Ln492-494)

c). Yes. The Matlab program automatically detects the border of the blood vessel by using the Gaussian process approach as mentioned above.

3) According to the description in the methods the authors do not represent calcium changes in percent (%) as stated, but simply by $\Delta F/F$ which lacks any unit. One can give these figures in % by multiplying them with 100, i.e. $\Delta F/F$ of 3 reflects a change in fluorescence of 300 %. This should be changed throughout the manuscript and in the figures (labeling of axes).

Thank you for bringing this to our attention. Calcium changes in this study were calculated by $\Delta F/F = [(F_{(t)} - F_{\text{baseline}}) / F_{\text{baseline}} * 100] \%$. The calcium changes shown in the figures have already been multiplied by 100, so we corrected the figures to indicate the units are percent (%) (Pg19, Ln487)

Results and conclusions

4) Fig. 5b shows a clear correlation between neuronal calcium and tissue oxygenation. The authors conclude that „cocaine-induced reduction in tissue oxygen content in the peak response period was due to increase in neuronal activation“ (p. 9, l. 196). A correlation does not indicate a causal connection. Indeed, the causal connection could be vice versa, with an increase in neuronal activation due to a reduction in tissue oxygenation and hence metabolic stress. The authors should change the manuscript accordingly to avoid any premature conclusions.

We agree. The sentence has been modified as

‘Though our findings cannot establish causality they suggests that cocaine-induced reduction in tissue oxygen content in the peak response period **might reflect** increased neuronal activation’

5) Fig. 6b. The same applies for the correlation between astrocytic calcium and total blood volume in arteries. The authors state that „astrocytic activation contributed to cocaine-induced vasoconstriction“ (p. 10, l. 213). Again, the correlation does not prove a causal connection.

We agree. The sentence has been adjusted as

“Though causality can not be established from our findings, results suggest that astrocytic activation by cocaine **underly cocaine induced** vasoconstriction”

6) **a)** The authors used calcium imaging to visualize astrocyte activation. The mechanism of cocaine-evoked calcium signaling in astrocytes is not clear, but it is likely that it involves calcium release from internal stores via the Gq-PLC pathway. One action of cocaine is inhibition of dopamine uptake, and the accumulated extracellular dopamine could activate astrocytic dopamine receptors and trigger Gq-coupled calcium release (Jennings et al. *Glia*. 2017 Mar;65(3):447-459. doi: 10.1002/glia.23103; Fischer et al. *Sci Rep*. 2020 Jan 20;10(1):631. doi: 10.1038/s41598-020-57462-4). Therefore, it is not clear why Gi-DREADD would inhibit astrocyte calcium signaling since Gi targets adenylate cyclase, not PLC. While it is shown that „inhibitory“ Gi-DREADD reduce excitation of neurons, the effect of Gi-DREADD and hence adenylate cyclase inhibition in astrocytes is barely studied and it can not be considered as evidenced that Gi-DREADD indeed inhibit astrocyte activation. **b)** I recommend that the authors perform control experiments to show whether Gi-DREADD activation suppresses cocaine-evoked calcium signaling in astrocytes.

a) Accumulating evidence indicates that drug exposures can have dynamic and long-lasting effects on glial cells including astrocytes.⁵ However, the mechanism underlying the effects of cocaine on vasoconstriction as well as those underlying astrocytic $[Ca^{2+}]_i$ increases remain largely unclear. Cocaine’s vasoconstriction effects are likely to reflect its sympathomimetic effects but its effects on L-type Ca channel’s function in blood vessels are also likely to contribute⁶. Astrocytic $[Ca^{2+}]_i$ accumulation associated with dopamine signaling involves Ca^{2+} release from internal stores via the Gq-PLC pathway^{7,8}. Also, the ionotropic receptors as well voltage-gated Ca^{2+} channels mediate Ca^{2+} influx into astrocytes⁹ that are involved in cocaine addiction¹⁰. Recently, our animal study¹¹ showed that Ca^{2+} channel blockade reduced cocaine’s vasoconstriction and neurotoxicity in the prefrontal cortex. Others reported that Ca^{2+} -channel blockers can also reduce negative outcomes from cocaine-induced cerebral ischemia and stroke by buffering cocaine-induced vasoconstriction¹², thus indicating the involvement of ionic homeostasis in vasoconstriction induced by cocaine. Additionally, cocaine triggers neuroadaptations^{10,13} in glutamate neurotransmission regulated by astrocytes that could further

worsen cocaine induced neurotoxicity. These complex roles of astrocytes in brain under cocaine-induced pathophysiological condition requires further investigation.

Fig.r6 Cocaine-induced astrocytic Ca^{2+}_A increase (a) and HbT decrease (b) were ameliorated by clozapine injection to active DREADD (Gi) signaling (a', b'). c,d) Quantification of cocaine-induced astrocytic Ca^{2+} fluorescence changes and HbT before (indicate in grey shadow) and after (indicate in pink shadow) astrocytic Gi signaling be active. Yellow shadow indicate significant difference ($p<0.05$) time period compare to baseline, and green shadow indicate no-significant time period ($p>0.05$).

b) **Fig.r6** shows the results from studies done to compare astrocytic Ca^{2+}_A dependent fluorescence and hemodynamic responses to cocaine in cortex of mice ($n=4$) before and after DREADD(Gi) activation by clozapine. Persistent HbT decrease ($\Delta[HbT]$, Fig. r6b) reflecting cocaine's vasoconstriction effects along with the corresponding Ca^{2+}_A increase ($\Delta[Ca^{2+}_A]$, Fig. r6a). As shown in Fig.r6b', $\Delta[HbT]$ response to cocaine was significantly attenuated after DREADD(Gi) activation by clozapine injection 30min prior to cocaine administration (from $-9.668 \pm 0.4799\%/min$ to $-0.6927 \pm 0.0933\%/min$ ($P^*\leq 0.001$)). Comparison of the cocaine-induced integrative $\Delta F/F[Ca^{2+}_A]$ changes before (shadowed as grey color in Fig. r6a) and after (pink color in Fig. r6a') Ca^{2+}_A inhibition is summarized in Fig.r6c, showing significant attenuation of cocaine induced changes in $\Delta[Ca^{2+}_A]$ ($P^*=0.004$), consistent with our hypothesis that Ca^{2+}_A mediates cocaine-induced hemodynamic changes. Thus the manipulation of Ca^{2+}_A signaling minimized cocaine-induced neurovascular effects in the cortex.

7) Fig. 7. Why did the authors change the read-out of vasoconstriction in their last set of experiments (measurement of blood vessel diameter instead of deltaHbT)? There are no control experiments showing the response to cocaine without Gi-DREADD for the change in vessel diameter. Please provide those controls. Furthermore, what is the effect of Gi-DREADD on deltaHbT in arteries? Without those experiments the results are not conclusive and the study rather preliminary.

As shown in **Fig. r4** above, the cocaine-induced hemodynamic changes without Gi-DREADDs active are included. Also comparison of changes in $\Delta\phi$, ΔHbO_2 , ΔHbT without and with Gi-DREADDs were conducted as addressed in question #2 in Rev#1 above.

Terminology

8) The authors use the term „Ca²⁺ fluorescence“ throughout the manuscript, they state that (GCaMP6f) fluorescence „was from astrocytic Ca²⁺“ and describe the anti-GFP immunostaining of GCaMP6f in neurons and astrocytes, respectively, as „neuronal Ca²⁺“ and „astrocytic Ca²⁺“ (Fig. 1), and so on. Ca²⁺ is non-fluorescent, and any fluorescence the authors refer to derives from GCaMP6f, not Ca²⁺ itself. Hence, the authors should go through the manuscript and change the terminology to make clear what they are talking about. However, Ca²⁺-dependent changes in fluorescence can be considered as calcium signals. Thus, it makes a difference whether one refers to fluorescence or to Ca²⁺-dependent fluorescence changes.

Thanks for pointing this out. We now specify ‘neuronal/astrocytic Ca²⁺’ as “neuronal/astrocytic GCaMP6f-expressing Ca²⁺”, and modified the statement of ‘Ca²⁺ fluorescence changes’ to ‘Ca²⁺-dependent fluorescence changes’ in the revised manuscript’. (Pg4; Pg5, Ln120)

Fig.r7 Ex vivo fluorescence images of GCaMP6f expression in neurons (a-c): GCaMP6f expression in glial cells (d-f). (g-i) showing the high-magnification ex vivo fluorescence images from (d-f).

Supplements

9) Fig. S1: The correct lettering of the figure panels following f is g, h, i (not j, ...).

Thank you for bringing this to our attention. We have corrected the typo in g as shown above.

10) Fig. S1 (j-l): „... the arrowheads (show) the connection between the astrocytes ...“ There are no arrowheads in the figure.

Thanks and we have corrected the caption for Figure s1 (g-l): the high-magnification ex vivo fluorescence images from (d-f), where the arrows show the connection between the astrocytes, and the dash circles show the territory of astrocytes...

11) Fig. S1 (j-l): The legend states that the depicted images are magnifications from the images in d-f. I don't see structures in d-f that resemble those in j-l. Please indicate the region in d-f that is magnified in j-l, e.g. by a square as overlay.

The images were retaken for Fig. S1 d-f, and now magnification area is specified (see Fig. r7 above).

Title

12) Place "with" after "associated".

Thanks and we have corrected the title the manuscript accordingly.

Reviewer #3 (Remarks to the Author):

*In this manuscript, the authors used a multimodal imaging platform to simultaneously measure neuronal and astrocytic Ca²⁺ signals, tissue oxygenation and vascular hemodynamics in response to acute application of cocaine. The authors reported that while cocaine reduced cerebral total blood volume, it only temporally decreased tissue oxygenation that became overshoot after returned to baseline. The overshoot was observed despite a sustained drop in total blood volume. Interestingly, while cocaine caused a biphasic neuronal Ca²⁺ response, cocaine did not elicit the same biphasic Ca²⁺ response in astrocytes. It was also reported that inhibition of astrocytic activity with GFAP-DREADD (Gi) blocked cocaine-induced vasoconstriction. **This is an interesting study with novel findings that provide insights into the interactions between astrocytes and the vasculature in response to cocaine and perhaps NVC in general.** Below are my comments.*

1. The basal neuronal Ca²⁺ signals appeared to be much strong than that of astrocytic Ca²⁺ signals, would the authors care to elaborate on the difference a bit? Would this affect the differences seen in cocaine-induced responses between the 2 cell types?

This is a great question. The astrocytic Ca²⁺ fluorescent changes in response to cocaine (i.e., $\Delta F/F_A = 2.106 \pm 0.4333\%$, n=5) is lower than the cocaine-induced neuronal Ca²⁺-dependent fluorescent changes ($\Delta F/F_N = 6.065 \pm 1.463\%$, n=5) as shown in Fig. 4 in the manuscript. To minimize the expression difference between these two groups of animals, we kept an identical volume of viral delivery into the cortex of all animals. Our ex vivo experiments indicated that there were no significant differences in GCaMP6f expression into neurons and astrocytes (n=3 animals/per group, ROIs=5/animal, p=0.94) shown in Fig.r7j above (or Fig s1j in revised manuscript). In addition, Ca²⁺-dependent fluorescent change was quantified as percent change relative to baseline. In other words, the effect of potential baseline variation between animals on the fluorescent signal are eliminated. Taken together, the amplitudes of fluorescent changes in G(N) and G(A) animals should represent the cocaine-induced intracellular calcium changes in neuron and astrocytes, respectively. Indeed, in our previous study we detected the cortical

Ca^{2+}_N and Ca^{2+}_A responses cortex to sensory stimulation in vivo, we found the Ca^{2+}_N fluorescent transients were stronger ($\Delta F/F_N=6.4\pm 0.29\%$) than Ca^{2+}_A fluorescent transients ($\Delta F/F_A=1.7\pm 0.1\%$), supporting that the potential difference in cellular Ca responses to the stimuli¹⁴.

We have discussed it in the revised manuscript (Pg15, Ln357-370)

2. It would be more informative if the authors could show the astrocytic Ca^{2+} signals in the GFAP-DREADD experiments.

We have taken this suggestion and run new animal experiments, in which the astrocytic Ca^{2+} signals were recorded as shown in Fig.r6 above.

These new experiments have been included in Fig.r4 and Fig.r6 above, which we put together to form a new Fig. 7 in the revised manuscript.

3. Did application of Clozapine affect the vessel diameter? 4. Did application of Clozapine affect neuronal Ca^{2+} ?

Fig.r8. Quantification of vessel diameter changes (purple) and neuronal Ca^{2+} (green) in response to clozapine (0.1mg/kg, i.p., n=3).

injection ($p=0.339$). Figure r8c shows the time traces of the vessel size change as a function of time in response to clozapine. One-way repeated ANOVA showed no significant time effect on vessel size after clozapine injection ($n = 3$). Quantification analysis of average efficiency shows that there is no significant difference before ($0.2778\% \pm 0.4723\%$) and after ($2.829\% \pm 1.458\%$) clozapine injection ($P=0.171$).

We have included the Fig.r8 in the supplemental figure S4.

To address these questions, we conducted additional experiments to inject clozapine (0.1mg/kg, i.p) and track the changes in the neuronal Ca^{2+} fluorescence and vessels in baseline (10 mins) and 30 mins after clozapine injection,

Our experimental results are summarized in Fig. r8 below. Fig. r8a shows the neuronal Ca^{2+} -dependent fluorescence changes as a function of time in response to clozapine. One-way repeated ANOVA showed no significant time effect on neuronal Ca^{2+} after clozapine injection ($n = 3$). Meanwhile, quantification analysis of average efficiency in Fig. r8b shows that there were significant differences before ($0.2145\% \pm 0.1361\%$) and after ($0.3940\% \pm 0.0940\%$) clozapine

Reference list for this reply

- 1 Han, K. *et al.* Neurovascular Coupling under Chronic Stress Is Modified by Altered GABAergic Interneuron Activity. *Journal of Neuroscience* **39**, 10081-10095 (2019).
- 2 Daneman, R. & Prat, A. The blood-brain barrier. *Cold Spring Harbor perspectives in biology* **7**, a020412, doi:10.1101/cshperspect.a020412 (2015).
- 3 Vecino, E., Rodriguez, F. D., Ruzafa, N., Pereiro, X. & Sharma, S. C. Glia-neuron interactions in the mammalian retina. *Progress in retinal and eye research* **51**, 1-40, doi:10.1016/j.preteyeres.2015.06.003 (2016).
- 4 Asl, M. E., Koohbanani, N. A., Frangi, A. F. & Gooya, A. Tracking and diameter estimation of retinal vessels using Gaussian process and Radon transform. *Journal of medical imaging (Bellingham, Wash.)* **4**, 034006, doi:10.1117/1.jmi.4.3.034006 (2017).
- 5 Reissner, K. J. & Pletnikov, M. V. Contributions of nonneuronal brain cells in substance use disorders. *Neuropsychopharmacol* **45**, 224-225, doi:10.1038/s41386-019-0494-5 (2020).
- 6 Addy, N. A. *et al.* The L-type calcium channel blocker, isradipine, attenuates cue-induced cocaine-seeking by enhancing dopaminergic activity in the ventral tegmental area to nucleus accumbens pathway. *Neuropsychopharmacol* **43**, 2361-2372 (2018).
- 7 Jennings, A. *et al.* Dopamine Elevates and Lowers Astroglial Ca²⁺ Through Distinct Pathways Depending on Local Synaptic Circuitry. *Glia* **65**, 447-459 (2017).
- 8 Wang, W., Shen, J., Lu, X., Hoi, S. C. H. & Ling, H. Paying Attention to Video Object Segmentation. *. IEEE TRANSACTIONS ON PATTERN ANALYSIS AND MACHINE INTELLIGENCE* **43(7):2413-2428** (2021).
- 9 Verkhatsky, A. & Nedergaard, M. Astroglial cradle in the life of the synapse. *Philosophical transactions of the Royal Society of London. Series B, Biological sciences* **369**, 20130595, doi:10.1098/rstb.2013.0595 (2014).
- 10 Wang, J. S. *et al.* Astrocytes in cocaine addiction and beyond. *Molecular psychiatry* **27**, 652-668 (2022).
- 11 Du, C. & Park, K. Ca²⁺ channel blockade reduces cocaine's vasoconstriction and neurotoxicity in the prefrontal cortex. **11**, 459, doi:10.1038/s41398-021-01573-7 (2021).
- 12 Kosten, T. R. Pharmacotherapy of cerebral ischemia in cocaine dependence. *Drug Alcohol Depen* **49**, 133-144 (1998).
- 13 D'Souza, M. S. Glutamatergic transmission in drug reward: implications for drug addiction. *Front Neurosci-Switz* **9** (2015).
- 14 Gu, X. *et al.* Synchronized Astrocytic Ca²⁺ Responses in Neurovascular Coupling during Somatosensory Stimulation and for the Resting State. *Cell reports* **23**, 3878-3890, doi:10.1016/j.celrep.2018.05.091 (2018).

REVIEWERS' COMMENTS:

Reviewer #1 (Remarks to the Author):

The authors have spent substantial effort in revising the manuscript and they have addressed all my concerns. Nice work!

Just one more suggestion that the figures prepared in the response letter along with the interpretation text shall be put in a supplemental material so that readers can have a better understanding of this work.

Reviewer #2 (Remarks to the Author):

Revision 1: Cocaine's astrocytic but not neuronal effects are associated cerebrovascular vasoconstriction: In vivo optical imaging of mouse brain
Liu et al.

The authors have addressed all of my points raised in the first review. However, obviously I did not make my point about the terminology of Ca²⁺ changes, fluorescent changes etc. clear enough and hence in some instances the edited phrases appear to be even more confusing than in the first version of the manuscript. I will list those instances here with suggestions in order to make the text better understandable to the readers.

Change ...

- Figure legend 1d) "Green: neuronal GCaMP6f-expressing Ca²⁺" to "Green: GCaMP6f-expressing neurons"

- Figure legend 1d) "Green: astrocytic GCaMP6f-expressing Ca²⁺" to "Green: GCaMP6f-expressing astrocytes"

- line 120: "... GCaMP6f-expressing neuronal (d1) and astrocytic (d2) Ca²⁺ dependent fluorescence in ...". This is fixed tissue, the fluorescence is not Ca²⁺-dependent but reflects the anti-GFP antibody labeling. Change to "... GCaMP6f-expressing neurons (d1) and astrocytes (d2) visualized by anti-GFP staining in ..."

- line 175: "Neuronal $\Delta F/FCa^{2+}$ -G(N)" to "Neuronal Ca²⁺"

- line 176: "Astrocytic $\Delta F/FCa^{2+}$ -G(N)" to "Astrocytic Ca²⁺"

- line 206: "neuronal fluorescence $\Delta F/FCa^{2+}$ " to "neuronal Ca²⁺"

- line 211: "neuronal fluorescence $\Delta F/FCa^{2+}$ " to "neuronal Ca²⁺"

Reviewer #3 (Remarks to the Author):

The authors have addressed my concerns. I think the manuscript is suitable for publication.

Response to the Reviewer Comments

We are very pleased that the reviewers recognized the unique aspects of this work. We also would like to express our appreciations to the editors and reviewers for their thoughtful comments and further suggestions on our manuscript. We have carefully re-revised our manuscript accordingly as highlighted in red in the updated manuscript. Below are our point-to-point responses to the referee comments.

Reviewer #1:

*Just one more suggestion that the figures prepared in the response letter along with the interpretation text shall be put in a supplemental material so that readers can have a better understanding of this work. **Editorial Comment: We would also ask that Rebuttal Figures 1-8 be integrated into the Supplementary Information as Supplementary Figures; please be sure to cite/discuss them accordingly in the main text.***

Per request, we have made changes as follows:

Rebuttal Figures 1-2 has been integrated as Supplemental Figures 6-7 and discussed in Pg13, Ln 496-500.

Rebuttal Figure 3 has been included in Figure 2 in revised manuscript and discussed in Pg5, Ln 148-151 and 157-161.

Rebuttal Figure 4 has been included in Figure 7 in revised manuscript and discussed in Pg7, Ln 233-259.

Rebuttal Figure 5 has been integrated as Supplemental Figures 5 and cited in Pg11, line 434-435.

Rebuttal Figure 6 has been included in Figure 7 in revised manuscript and discussed in Pg7, line 233-259.

Rebuttal Figure 7 has been integrated as Supplemental Figures 1 and discussed in Pg9 , Ln 347-356.

Rebuttal Figure 8 has been integrated as Supplemental Figures 4 and discussed in Pg7, Ln260-273.

Reviewer #2:

The authors have addressed all of my points raised in the first review. However, obviously I did not make my point about the terminology of Ca²⁺ changes, fluorescent changes etc. clear enough and hence in some instances the edited phrases appear to be even more confusing than in the first version of the manuscript. I will list those instances here with suggestions in order to make the text better understandable to the readers.

Change ...

Figure legend 1d) "Green: neuronal GCaMP6f-expressing Ca²⁺" to "Green: GCaMP6f-expressing neurons". Figure legend 1d) "Green: astrocytic GCaMP6f-expressing Ca²⁺" to "Green: GCaMP6f-expressing astrocytes". line 120: "... GCaMP6f-expressing neuronal (d1) and astrocytic (d2) Ca²⁺ dependent fluorescence in ...". This is fixed tissue, the fluorescence is not Ca²⁺-dependent but reflects the anti-GFP antibody labeling. Change to "... GCaMP6f-expressing neurons (d1) and astrocytes (d2) visualized by anti-GFP staining in ...". line 175: "Neuronal $\Delta F/FCa^{2+}$ -G(N)" to "Neuronal Ca²⁺". line 176: "Astrocytic $\Delta F/FCa^{2+}$ -G(N)" to "Astrocytic Ca²⁺". line 206: "neuronal fluorescence $\Delta F/FCa^{2+}$ " to "neuronal Ca²⁺". line 211: "neuronal fluorescence $\Delta F/FCa^{2+}$ " to "neuronal Ca²⁺"

Thank you for the suggestion. We have modified the statement by following reviewer's suggestions in the revised manuscript and marked in red color.